# Lattice quantum electrodynamics in (3+1)-dimensions at finite density with tensor networks

Giuseppe Magnifico [1,2✉], Timo Felser [1,2,3], Pietro Silvi[4,5] & Simone Montangero [1,2]

Gauge theories are of paramount importance in our understanding of fundamental constituents of matter and their interactions. However, the complete characterization of their phase diagrams and the full understanding of non-perturbative effects are still debated, especially at finite charge density, mostly due to the sign-problem affecting Monte Carlo numerical simulations. Here, we report the Tensor Network simulation of a three dimensional lattice gauge theory in the Hamiltonian formulation including dynamical matter: Using this sign-problem-free method, we simulate the ground states of a compact Quantum Electrodynamics at zero and finite charge densities, and address fundamental questions such as the characterization of collective phases of the model, the presence of a confining phase at large gauge coupling, and the study of charge-screening effects.

[1] Dipartimento di Fisica e Astronomia G. Galilei, Università di Padova, Padova, Italy. [2] Istituto Nazionale di Fisica Nucleare (INFN), Sezione di Padova, Padova, Italy. [3] Theoretische Physik, Universität des Saarlandes, Saarbrücken, Germany. [4] Center for Quantum Physics, Institute for Experimental Physics, University of Innsbruck, Innsbruck, Austria. [5] Institute for Quantum Optics and Quantum Information, Austrian Academy of Sciences, Innsbruck, Austria. ✉email: giuseppe.magnifico@unipd.it

anging from high-energy particle physics (Standard Model)[1–3] to low-temperature condensed matter physics (spin liquids, quantum Hall, and high-$T_c$ superconductivity)[4,5], gauge theories constitute the baseline in our microscopical description of the universe and are a cornerstone of contemporary scientific research. Yet, capturing their many-body body behavior beyond perturbative regimes, a mandatory step before experimentally validating these theories often eludes us[6]. One for all, the quark confinement mechanism in quantum chromodynamics, a founding pillar of the Standard Model which has been studied for almost half a century, is still at the center of current research efforts[7–12]. Indeed, a powerful numerical workhorse such as Monte-Carlo simulations[13–15], capable of addressing discretized lattice formulations of gauge theories[11,16–18], struggles in highly interesting regimes, where matter fermions and excess of charge are concerned, due to the infamous sign problem[19]. In recent years, a complementary numerical approach, Tensor Networks (TN) methods, have found increasing applications for studying low-dimensional Lattice Gauge Theories (LGT) in the Hamiltonian formulation[20,21]. As tailored many-body quantum state ansätze, TNs are an efficient approximate entanglement-based representation of physical states, capable of efficiently describe equilibrium properties and real-time dynamics of systems described by complex actions, where Monte Carlo simulations fail to efficiently converge[22]. TN methods have proven remarkable success in simulating LGTs in (1+1) dimensions[23–41], and very recently they have shown potential in (2+1) dimensions[42–50]. To date, due to the lack of efficient numerical algorithms to describe high-dimensional systems via TNs, no results are available regarding the realistic scenario of LGTs in three spatial dimensions.

In this work, we bridge this gap by numerically simulating, via TN ansatz states, an Abelian lattice gauge theory akin to (3 + 1) Quantum Electrodynamics (QED), at zero temperature. We show that, by using the quantum link formalism (QLM) of LGTs[51,52] and an unconstrained Tree TN (TTN), we can access multiple equilibrium regimes of the model, including finite charge densities. Precisely, we analyze the ground state properties of quantum-link QED in (3 + 1)D for intermediate system sizes, up to 512 lattice sites. The matter is discretized as a staggered spinless fermion field on a cubic lattice[16], while the electromagnetic gauge fields are represented on lattice links, and truncated to a compact representation of spin-$s$. Here, we present results from a nontrivial representation for lattice gauge fields (the spin-1 case), with possible generalizations to higher spin requiring only a polynomial overhead in $s$. Our picture can be similarly adapted to embed non-Abelian gauge symmetries, such as they appear in QCD[32]. Finally, we stress that the truncation of the gauge field is a common step in quantum simulations and computations[53–62], making the presented numerical approach a landmark benchmarking and cross-verification tool for current and future experiments. By variationally approximating the lattice QED ground state with a TTN, we address a variety of regimes and questions inaccessible before. In the scenario with zero excess charge density, we observe that the transition between the vacuum phase and the charge-crystal phase is compatible with a second-order quantum phase transition[47]. In the limit of zero magnetic couplings, this transition occurs at negative bare masses $m_0$, but as the coupling is activated, the critical point is shifted to larger, and even positive, $m_0$ values. To investigate field-screening properties, we also consider the case where two parallel charged plates are placed at a distance (a capacitor). By studying the polarization of the vacuum in the inner volume, we observe an equilibrium string-breaking effect akin to the Schwinger mechanism. Furthermore, we address the confinement problem by evaluating the binding energies of charged particle pairs pinned at specified distances. Finally, we consider the scenario with a charge imbalance into the system, i.e., at finite charge density, and we characterize a regime where charges accumulate at the surface of our finite sample, analogously to a classic perfect conductor.

## Results

**The model.** Hereafter, we numerically simulate, at zero temperature, the Hamiltonian of $U(1)$ quantum electrodynamics on a finite $L \times L \times L$ three-dimensional simple cubic lattice[16–18]

$$\hat{H} = -t\sum_{\mathbf{x},\mu}\left(\hat{\psi}_{\mathbf{x}}^{\dagger}\,\hat{U}_{\mathbf{x},\mu}\,\hat{\psi}_{\mathbf{x}+\mu}+\text{H.c.}\right) \tag{1a}$$

$$+m\sum_{\mathbf{x}}(-1)^{\mathbf{x}}\hat{\psi}_{\mathbf{x}}^{\dagger}\hat{\psi}_{\mathbf{x}}+\frac{g_{\mathrm{e}}^2}{2}\sum_{\mathbf{x},\mu}\hat{E}_{\mathbf{x},\mu}^2 \tag{1b}$$

$$-\frac{g_{\mathrm{m}}^2}{2}\sum_{x}\left(\hat{\square}_{\mu_x,\mu_y}+\hat{\square}_{\mu_x,\mu_z}+\hat{\square}_{\mu_y,\mu_z}+\text{H.c.}\right) \tag{1c}$$

with $\mathbf{x} \equiv (i,j,k)$ for $0 \le i,j,k \le L-1$ labeling the sites of the lattice and $\hat{\square}_{\mu_\alpha,\mu_\beta} = \hat{U}_{\mathbf{x},\mu_\alpha}\hat{U}_{\mathbf{x}+\mu_\alpha,\mu_\beta}\hat{U}_{\mathbf{x}+\mu_\beta,\mu_\alpha}^{\dagger}\hat{U}_{\mathbf{x},\mu_\beta}^{\dagger}$. Here, we adopted the Kogut–Susskind formulation[16], representing fermionic degrees of freedom with a staggered spinless fermion field $\{\hat{\psi}_{\mathbf{x}},\hat{\psi}_{\mathbf{x}'}^{\dagger}\} = \delta_{\mathbf{x},\mathbf{x}'}$ on lattice sites. Their bare mass $m_{\mathbf{x}} = (-1)^{\mathbf{x}}m$ is staggered, as tracked by the site parity $(-1)^{\mathbf{x}} = (-1)^{i+j+k}$, so that fermions on even sites represent particles with positive electric charge $+q$, while holes on odd sites represent anti-particles with a negative charge $-q$, as shown in Fig. 1. Charge $\hat{Q}$ conservation is thus expressed as global fermion number $\hat{N}$ conservation, since $\hat{Q} = \sum_{\mathbf{x}}\left(\hat{\psi}_{\mathbf{x}}^{\dagger}\hat{\psi}_{\mathbf{x}}-\frac{1-(-1)^{\mathbf{x}}}{2}\right) = \hat{N}-L^3/2$.

The links of the 3D lattice are uniquely identified by a couple of parameters $(\mathbf{x},\mu)$ where $\mathbf{x}$ is any site, $\mu$ is one of the three positive lattice unit vectors $\mu_x \equiv (1,0,0)$, $\mu_y \equiv (0,1,0)$, and $\mu_z \equiv (0,0,1)$. The gauge fields are defined on lattice links through the pair of operators $\hat{E}_{\mathbf{x},\mu}$ (electric field) and $\hat{U}_{\mathbf{x},\mu}$ (unitary comparator) that satisfy the commutation relation

$$[\hat{E}_{\mathbf{x},\mu},\hat{U}_{\mathbf{x}',\mu'}] = \delta_{\mathbf{x},\mathbf{x}'}\delta_{\mu,\mu'}\hat{U}_{\mathbf{x},\mu'}. \tag{2}$$

For comfort of notation, we can extend the definition to negative lattice unit vectors via $\hat{E}_{\mathbf{x}+\mu,-\mu} = -\hat{E}_{\mathbf{x},\mu}$ and $\hat{U}_{\mathbf{x}+\mu,-\mu} = \hat{U}_{\mathbf{x},\mu}^{\dagger}$.

The Hamiltonian of Eqs. (1a)–(1c) consists of four terms: the parallel transporter (1a) describes the creation and annihilation of a particle–antiparticle pair, shifting the gauge field in-between to preserve local gauge symmetries. The staggered mass and the electric energy density (1b) are completely local. Finally, the plaquette terms (1c) capture the magnetic energy density and are related to the smallest Wilson loops along the closed plaquettes along the three planes x–y, x–z, y–z of the lattice. In dimensionless units ($\hbar = c = 1$), the couplings in Eqs. (1a)–(1c) are not independent: They can be expressed as $t = 1/a$, $m = m_0$, $g_{\mathrm{e}}^2 = g^2/a$, $g_{\mathrm{m}}^2 = 8/(g^2a)$, where $a$ is the lattice spacing, $g$ is the coupling constant of QED, and $m_0$ is the bare mass of particles/antiparticles. The numerical setup allows us to consider the couplings ($t, m, g_{\mathrm{e}}, g_{\mathrm{m}}$) as mutually independent. We then recover the physical regime of QED by enforcing $g_{\mathrm{e}}g_{\mathrm{m}} = 2\sqrt{2}t$[16]. We also fix the energy scale by setting $t = 1$.

The local $U(1)$ gauge symmetry of the theory is encoded in Gauss's law, whose generators

$$\hat{G}_{\mathbf{x}} = \hat{\psi}_{\mathbf{x}}^{\dagger}\hat{\psi}_{\mathbf{x}}-\frac{1-(-1)^{\mathbf{x}}}{2}-\sum_{\mu}\hat{E}_{\mathbf{x},\mu}, \tag{3}$$

are defined around each lattice site $\mathbf{x}$. The sum in Eq. (3) involves the six electric field operators on the links identified by $\pm\mu_x$, $\pm\mu_y$,

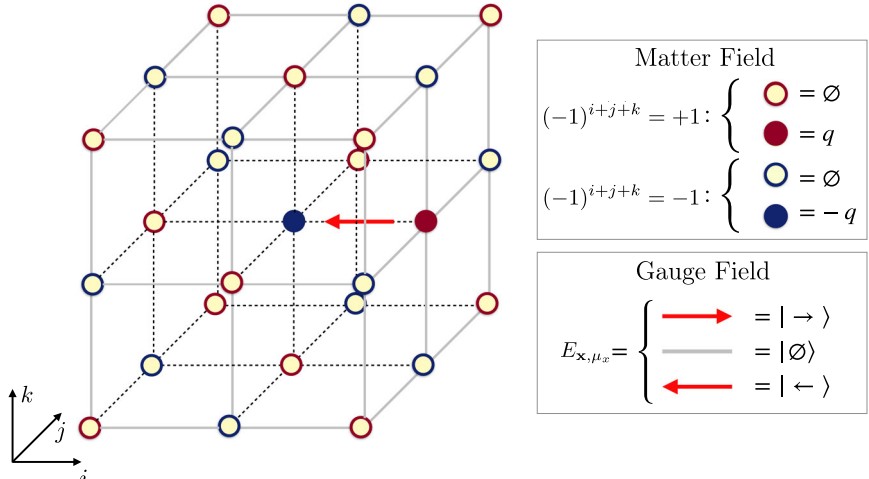

**Fig. 1 Scheme of the three-dimensional LGT with three electric field levels (spin-1 compact representation).** Fermionic degrees of freedom are represented by staggered fermions on sites with different parity: on the even (odd) sites, a full red (blue) circle corresponds to a particle (antiparticle) with a positive (negative) charge. As an illustrative example, it is shown a gauge-invariant configuration of matter and gauge fields with one particle and one antiparticle in the sector of zero total charge.

$\pm\mu_z$. Each $\hat{G}_\mathbf{x}$ commutes with the Hamiltonian $\hat{H}$. In the absence of static (background) charges, the gauge-invariant Hilbert space consists of physical many-body quantum states $|\Phi\rangle$ satisfying $\hat{G}_\mathbf{x}|\Phi\rangle = 0$ at every site $\mathbf{x}$.

As stressed in the standard Wilson's formulation of lattice QED[11], faithful representations of the $(\hat{\mathbf{E}}, \hat{\mathbf{U}})$ algebra are infinite-dimensional. A truncation to a finite dimension becomes therefore necessary for numerical simulations with TN methods, which require a finite effective Hilbert dimension at each lattice site. We use the quantum link model (QLM) approach in which the gauge field algebra is replaced by $SU(2)$ spin algebra, i.e., $\hat{E}_{\mathbf{x},\mu} \equiv \hat{S}^z_{\mathbf{x},\mu}$ and $\hat{U}_{\mathbf{x},\mu} \equiv \hat{S}^+_{\mathbf{x},\mu}/s$ for a spin-$s$ representation. This substitution keeps the electric field operator hermitian and preserves Eq. (2), but $\hat{U}$ is no longer unitary. Throughout this work, we will select $s = 1$, the smallest representation ensuring a nontrivial contribution of all the terms in the Hamiltonian (see also Fig. 1). This truncation introduces a local energy cutoff based on $g_e^2$, which in turn requires larger spin $s$ to accurately represent weaker coupling regimes, still potentially accessible via TNs[31].

**Transition at zero charge**. We focus on the zero charge sector, i.e., $\sum_\mathbf{x} \hat{\psi}^\dagger_\mathbf{x}\hat{\psi}_\mathbf{x} = \frac{L^3}{2}$, and periodic boundary conditions. As shown in Fig. 2 (upper panel), for $g_m^2 = 0$ the system undergoes a transition between two regimes, analogously to the $(1+1)$D and $(2+1)$D cases[25,37,47]: for large positive masses, the system approaches the bare vacuum, while for large negative masses, the system is arranged into a crystal of charges, a highly degenerate state in the semiclassical limit $(t \to 0)$ due to the exponential number of electric field configurations allowed. We track this transition by monitoring the average matter density $\rho = \frac{1}{L^3}\sum_\mathbf{x}<\text{GS}|\hat{n}_\mathbf{x}|\text{GS}>$, where $\hat{n}_\mathbf{x} \equiv \frac{1+(-1)^\mathbf{x}}{2} - (-1)^\mathbf{x}\hat{\psi}^\dagger_\mathbf{x}\hat{\psi}_\mathbf{x}$ is the matter occupation operator and the many-body ground state $|\text{GS}>$ has been computed by TTN algorithm (see the "Methods" section for details). Figure 2b displays the result for different sizes $L$ (and $g_e^2/2 = t = 1$), portraying the transition. Panels (a) and (c) display local configurations of matter $<\hat{n}_\mathbf{x}>$ and gauge sites $\langle\hat{E}_{\mathbf{x},\mu}\rangle$ for $m = -3.0$ and $m = +3.0$ respectively. In the former regime, the algorithm seems to favor a single allowed configuration of gauge fields rather than a superposition of many configurations: This is due to the fact that, when $g_m^2 = 0$, the matrix element that

rearranges the configurations occurs at very high perturbative order in $|t/m|$, and is numerically neglected. A finite-size scaling analysis of the transition (as detailed in the Methods' subsection "Critical points: scaling analysis") yields results compatible with a II-order phase transition, with the critical point occurring at negative bare masses $m$.

The same transition appears to be more interesting when we activate the magnetic coupling, by setting $g_m^2 = 8t^2/g_e^2 = 4$ (physical line). The phase at large negative $m$ now appears to be a genuine superposition of many configurations of the electric field, as they are coupled by matrix elements of the order $\sim g_m^2$, kept as numerically relevant by the algorithm. Moreover, the transition is still compatible with an II-order phase transition, and the critical point is shifted to larger $m$ values. This can lead to a critical bare mass $m_c$ that is positive (as we observed $m_c \approx +0.22$ for the case $g_e^2/2 = t = 1$), ultimately making the transition physically relevant.

**Quantum capacitor**. To investigate field-screening and equilibrium string-breaking properties, we analyze the scenario where two charged plates (an electric capacitor) are placed at the opposite faces of a volume, with open boundary conditions (OBC). In our simulations, we achieve this regime by setting large local chemical potentials on the two boundaries. We expect that for small positive masses $m$, the vacuum inside the plates will spontaneously polarize to an effective dielectric, by creating particle and antiparticle pairs to screen the electric field from the plates, into an energetically favorable configuration.

We observe this phenomenon by monitoring the charge density function along the direction $\mu_x$ orthogonal to the plates $q_c(d) = \frac{2}{L^2}\sum_{j,k=1}^L <\text{GS}|(-1)^\mathbf{x}\hat{\psi}^\dagger_{(d,j,k)}\hat{\psi}_{(d,j,k)}|\text{GS}>$ as well as the electric field amplitude along $\mu_x$, $E^c_{\mathbf{x},\mathbf{x}+\mu_x}(d) = \frac{2}{L^2}\sum_{j,k=1}^L <\text{GS}|\hat{E}_{(d,j,k),(d+1,j,k)}|\text{GS}>$, as presented in Fig. 3.

A transition from a vacuum regime to a string-breaking dielectric regime is observed, when driving $m$ from negative to positive. However, here the critical point occurs at positive masses $(m_c > 0)$ even at zero magnetic coupling $g_m^2 = 0$, analogously to the $(1+1)$D case[25]. In conclusion, the charged capacitor can make the phase transition physical even when $g$ can not be tuned.

The observed behavior can be interpreted as an equilibrium counterpart to the *Schwinger mechanism*, a real-time dynamical

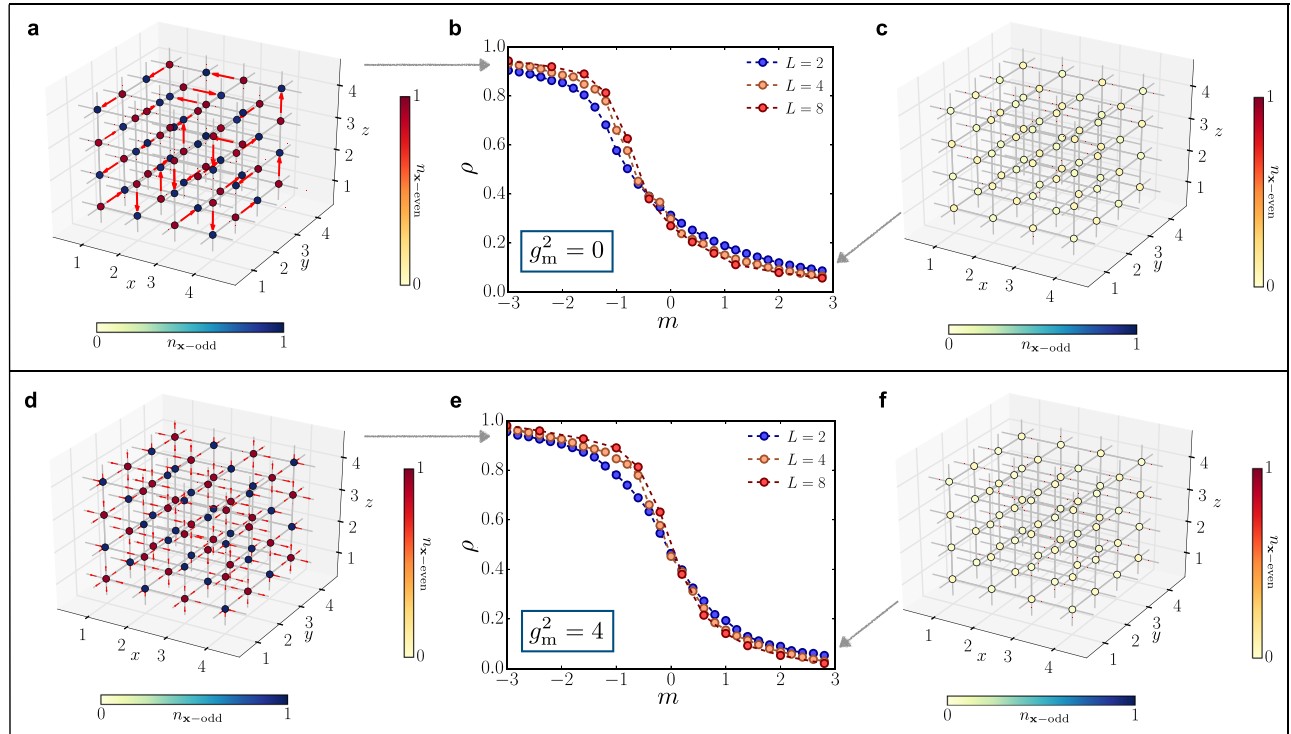

**Fig. 2 Transition at zero total charge.** Ground state charge occupation and electric field on links for $m = -3.0$ (**a**) and $m = 3.0$ (**c**) and $g_m^2 = 0$. **b** Particle density as a function of $m$, for different system size $L$ and $g_m^2 = 0$. Ground state charge occupation and electric field on links for $m = -3.0$ (**d**) and $m = 3.0$ (**f**) in the presence of magnetic interactions with $g_m^2 = 8/g_e^2 = 4$. **e** Particle density as a function of $m$, for different system size $L$ and $g_m^2 = 8/g_e^2 = 4$.

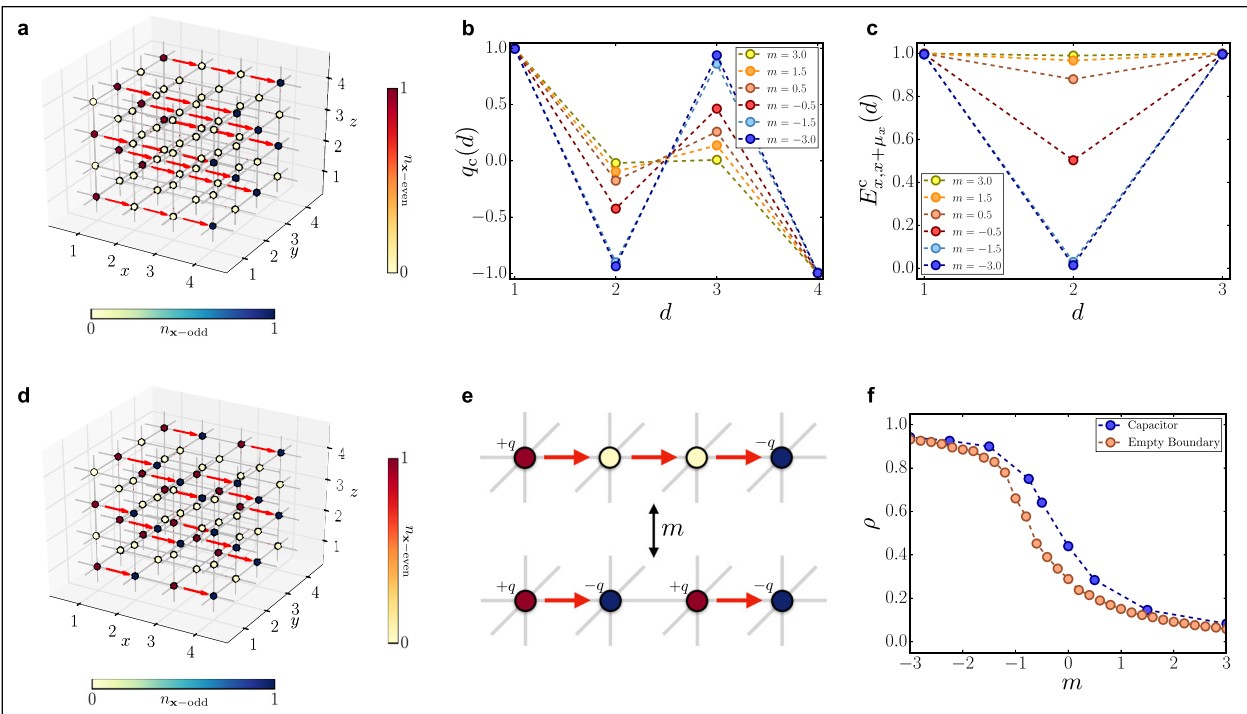

**Fig. 3 Quantum capacitor properties. a** Ground state configuration of the quantum capacitor for $m = 3.0$. **b** Mean charge density on the sites along the transverse direction for different values of $m$. **c** Mean value of the electric field on the transverse links for different values of $m$. **d** Ground state configuration of the quantum capacitor for $m = -3.0$. **e** Illustration of the creation of a particle-antiparticle pair along the transverse direction, starting from the initial electric field string generated by the boundary charges. **f** Particle density as a function of $m$, with a comparison to the case with no boundary charges.

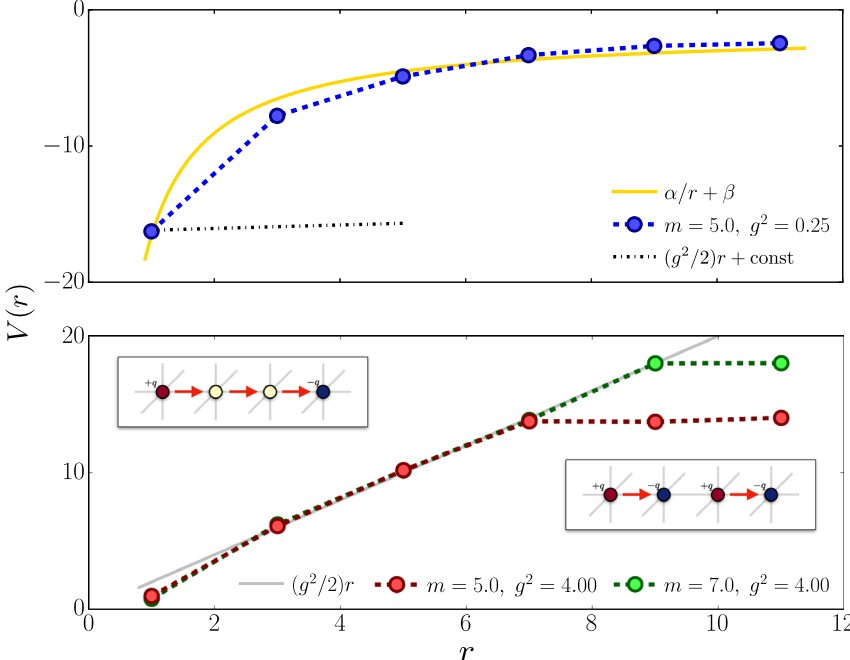

**Fig. 4 Confinement properties.** Interaction potential $V(r)$ between two charges of opposite sign as a function of their distance $r$ in the (upper panel) weak coupling regime $g \ll 1$ and (lower panel) strong coupling regime $g \gg 1$.

phenomenon in which the spontaneous creation of electron-positron pairs out of the vacuum is stimulated by a strong external electric field[63]. This could either be potentially verified in experiments or quantum simulations, by means of adiabatic quenches, ramping up the capacitor voltage.

**Confinement properties.** The $(3 + 1)$-dimensional pure compact lattice QED predicts a confining phase at large coupling $g$[11,64–67]. This phase, where the magnetic coupling is negligible, is characterized by the presence of a linear potential between static test charges, and is expected to survive at the continuum limit. By decreasing $g$, the system undergoes a phase transition to the Coulomb phase where the magnetic terms are not negligible and the static charges interact through the $1/r$ Coulomb potential at distance $r$[68]. When the gauge field is coupled to dynamical matter ($t \neq 0$ and finite $m$), new possible scenarios emerge, such as the string-breaking mechanism. Nevertheless, the transition between confined and deconfined phases is still expected to occur[69].

We can investigate this specific scenario with our TN method: we consider a $16 \times 4 \times 4$ lattice and pin two opposite charges via large local chemical potentials at distance $r$ along direction $\mu_x$. The energy $E(r) = V(r) - V(\infty) + 2\epsilon_1 + E_0$ of this ground state comprises: the work $V(r) - V(\infty)$ needed to bring two charges from infinity to distance $r$, plus twice the excitation energy $\epsilon_1$ of an isolated pinned charge, on top of the dressed-vacuum energy $E_0$. Therefore we can estimate the interaction potential as $V(r) = E(r) - E_0 + \xi$ where the additive constant $\xi$ does not scale with the volume (while $E(r)$ and $E_0$ separately do).

The presence of dynamical matter heavily impacts the strong-coupling picture ($g_m^2 \sim 0$), as it can be extrapolated in the semiclassical limit ($t \sim 0$). Here, a particle-antiparticle pair at distance $r$ with, a field-string between them, has an energy

$$E(r) - E_0 = 2m + \frac{g^2}{2a^2}\, r. \tag{4}$$

that scales linearly with $r$ (here $g^2 = a g_e^2$). By contrast, two mesons (neighboring particle–antiparticle pairs) have a flat

energy profile

$$E_{\text{pairs}} - E_0 = 4m + \frac{g^2}{a}. \tag{5}$$

Thus, for any mass $m$, there is critical distance $r_0$ above which the string is broken, and the formation of two mesons is energetically favorable.

We observe this transition at finite $t$, as shown in Fig. 4 (bottom panel, $g^2 = 4$). The crossover from the short-range to long-range behavior is still relatively sharp, and the distance $r_c$ at which it occurs strongly depends on the bare mass $m$. This is in contrast to the weak-coupling regime (top panel, $g^2 = 1/4$), where the potential profile $V(r)$ is smoothly increasing with $r$, and its slope at short distances disagrees with the string tension ansatz $rg^2/2 + \text{const.}$. Thus our simulations highlight visibly different features between confined and deconfined regimes, even with the dynamical matter.

**Finite density.** One of the most important features of our numerical approach is the possibility to tackle finite charge-density regimes. In fact, by exploiting the global $U(1)$ Fermion-number symmetry, implemented in our TTN algorithms, we can inject any desired charge imbalance into the system, while working under OBC. Figure 5 shows the results for charge density $\rho = Q/L^3 = 1/4$. In the vacuum phase ($m \gg g_e^2/2 \approx t$), we obtain configurations as displayed in panel (a), where the charges are expelled from the bulk and stick to the boundaries to minimize the electric field energy of the outcoming fields. To quantify this effect, which can also be interpreted as a field-screening phenomenon, we introduce the surface charge density

$$\sigma(l) = \frac{1}{A(l)} \sum_{\mathbf{x} \in A(l)} \left\langle \hat{\psi}_{\mathbf{x}}^\dagger \hat{\psi}_{\mathbf{x}} \right\rangle \tag{6}$$

where $A(l)$ contains only sites sitting at lattice distance $l$ from the closest boundary. The deeper we are in the vacuum phase, the faster the surface charge decays to zero away from the boundary

**a**

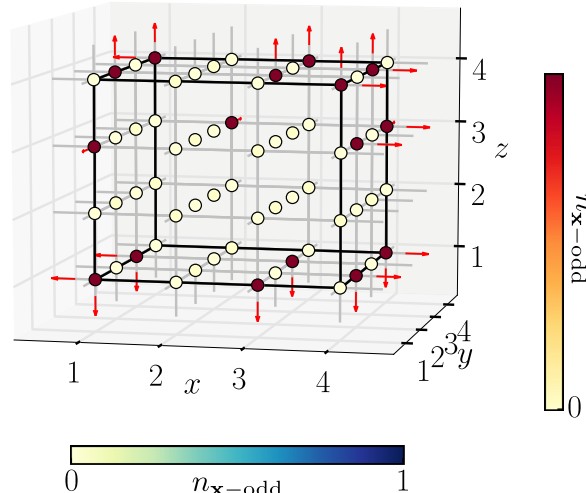

**b**

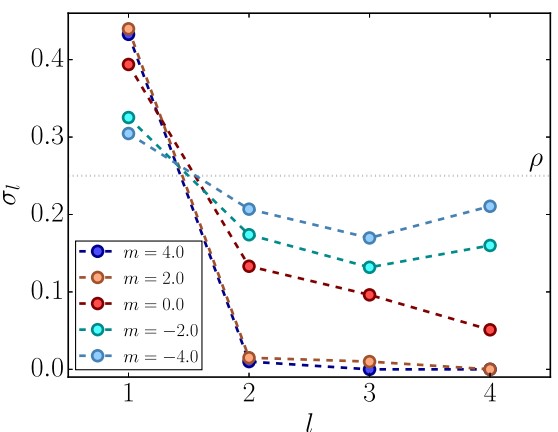

**Fig. 5 Finite density analysis. a** Ground state configuration for $m = 4.0$ at finite charge density $\rho = Q/L = 1/4$. The system is in the global symmetry sector with $Q = 16$ positive charges on the lattice with linear size $L = 4$. **b** Surface charge density $\sigma_l$ on a cube whose faces are at distance $l$ from the boundaries of the lattice with linear size $L = 8$. The system is in the global symmetry sector with $Q = 128$ positive charges (finite density $\rho = 1/4$).

($l = 1$). By contrast, close to the transition, the spontaneous creation of charge-anticharge pairs determines a finite charge density of the bulk. Finally, for large negative $m$, the charge distribution is roughly uniform.

## Discussion

We have shown that TN methods can simulate LGT in three spatial dimensions, in the presence of matter and charge imbalance, ultimately exploring those regimes where other known numerical strategies struggle. We have investigated collective phenomena of lattice QED which stand at the forefront of the current research efforts, including quantum phase diagrams, confinement issues, and the string breaking mechanism at equilibrium. We envision the possibility of including more sophisticated diagnostic tools, such as the 't Hooft operators[70] which nicely fit TNs designs, to provide more quantitatively precise answers to the aforementioned open problems.

From a theoretical standpoint, our work corroborates the long-term perspective to employ TN methods to efficiently tackle non-

perturbative phenomena of LGTs, in high dimensions and in regimes that are out of reach for other numerical techniques. As ansatz states with a refinement parameter chosen by the user, the bond dimension, TTNs are automatically equipped with a self-validation tool: convergence of each quantity with the bond dimension can be verified in polynomial time.

However, while TTNs perform well for small and intermediate system sizes, as the ones considered in this work ($L = 2, 4, 8$), the pathway to general LGTs analysis with large $L$ is still a technical challenge. In particular, TTNs suffer from poor scalability for higher $L$, since they fail to explicitly capture area law for large systems, which denotes a possible bottleneck for further investigations toward the study of the thermodynamical limit of Abelian and non-Abelian high-dimensional LGTs. As a promising perspective, ref. [71] presents the *augmented* TTN ansatz which compensates this drawback offering better scalability. Further development in this direction will contribute to overcoming the current limitations of TTN in high-dimensional systems opening the pathway to the possibility of investigating realistic physics by starting from the TTN approach presented here.

Furthermore, we stress that our simulations have been performed on standard clusters by taking advantage only of OpenMP parallelization on single multi-core nodes. We have not yet exploited a full-scale parallelization on multi-node architectures. At a purely technical level, it is straightforward to upgrade our algorithms in this direction, in order to fully exploit the capabilities of high-performance computing. For instance, following the ideas presented in ref. [72], each TTN variational sweep could be parallelized in a way to optimize its tensors separately on different computing nodes, so as to optimally scale the computational resources with the system size. On top of this, the implementation of tensor contractions on GPUs could be used to speed up the low-level computations as well[73].

In this work we consider the spin $s = 1$ representation, which leads to a local basis dimension of 267, as described in the Methods' subsection "Fermionic compact representation of local gauge-invariant site". Following the same theoretical construction for the local gauge-invariant sites, we estimate a local basis dimension of 1102 for the next representation of QED with $s = 3/2$, whereas, for the SU(2) Yang–Mills theory, by truncating after the first nontrivial irreducible representation and considering the spin-1/2 fermionic matter, one finds a local basis of 178 states for the cubic lattice. TTNs algorithms scale only polynomially with the local basis dimension but taking into account the aforementioned numbers, specific strategies for truncating also the local dimension in an optimal way (see for instance ref. [74]) could be required for studying higher representations of the gauge fields.

In conclusion, the aforementioned technical steps will be fundamental to tackle the problem of the continuum limit of realistic Abelian and non-Abelian LTGs and we foresee that, although very challenging, they are only some steps away along the path of TTNs developments presented here.

Alongside, from an experimental point of view, QLM formulations are among the most studied pathway towards the simulation of LGTs on quantum hardware[61,75]. The recent developments in low-temperature physics and control techniques, for trapped ions, ultracold atoms, and Rydberg atoms in optical lattices, have led to the first experimental quantum simulations of one-dimensional LGTs[55–59]. In this framework, numerical methods capable of accessing intermediate sizes, such as TNs, play a fundamental role as a cross-verification toolbox.

## Methods

**Fermionic compact representation of the local gauge-invariant site.** In describing a framework for LGT, a common requirement of TN numerical

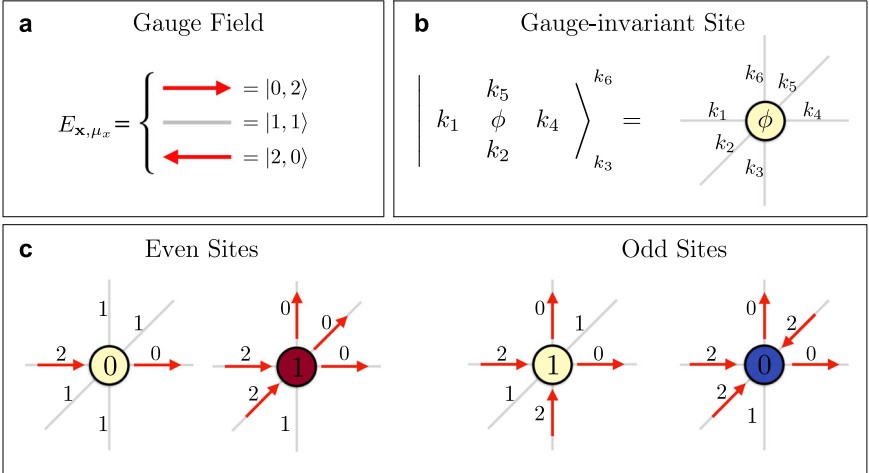

**Fig. 6 Construction of the gauge-invariant configurations for the local sites. a** Representation of the gauge field in terms of two species of Dirac modes in the sector with a total number of fermions equal to two. **b** Generic state of the local site composed by the matter degrees of freedom and six half-links along with the three spatial directions. On each half-link, the coefficients $k_j \in \{0, 1, 2\}$ define the fermionic modes. **c** Examples of gauge-invariant configurations for even and odd sites. Due to the use of staggered-fermions, the presence/absence of a fermion in an even/odd site represents the presence of a charge/anti-charge.

simulations[76–79], as well as quantum simulations[80–86], is working with finite-dimensional local degrees of freedom. This is a hard requirement when investigating both LGT descending from high-energy quantum field theories[87–90], and condensed matter models with emergent gauge fields[91,92]. While other pathways have been developed[93–96], in this work we adopted the well-known approach of truncating the gauge field space based on an energy density cutoff. In this section, we present the construction of the QED gauge-invariant configurations for the local sites that we exploit as a computational basis in our TN algorithm.

The use of the spin-1 representation implies that the gauge degrees of freedom on each link of the lattice is represented by three orthogonal eigenstates of the electric field operator

$$\hat{E}_{\mathbf{x},\mu}|\rightarrow\rangle = |\rightarrow\rangle, \ \hat{E}_{\mathbf{x},\mu}|\phi\rangle = 0, \ \hat{E}_{\mathbf{x},\mu}|\leftarrow\rangle = -|\leftarrow\rangle. \tag{7}$$

The parallel transporter, which is proportional to the raising operator in the spin language, acts on these states as

$$\hat{U}_{\mathbf{x},\mu}|\rightarrow\rangle = 0, \ \hat{U}_{\mathbf{x},\mu}|\phi\rangle = |\rightarrow\rangle, \ \hat{U}_{\mathbf{x},\mu}|\leftarrow\rangle = |\phi\rangle. \tag{8}$$

In the following, in order to obtain a representation of the gauge degrees of freedom that will be useful for constructing our TN ansatz, we employ the local mapping presented in ref. [47] (see also[97,98]), generalizing it to the case with three spatial dimensions. This technique is related to the standard rishon formulation of QLM[99–101] and allows us to encode Gauss's law taking into account the anticommutation relations of the fermionic particles on the lattice.

Let us consider a generic link of the lattice $(\mathbf{x}, \mu)$ between the two sites $\mathbf{x}$ and $\mathbf{x} + \mu$: the starting point is the splitting of the gauge field of this link into a pair of *rishon* modes so that each mode belongs to either one of the two sites. For the $s = 1$ case, we can set each rishon mode (or half-link) to be a three-hardcore fermionic field $\hat{\eta}_{\mathbf{x},\mu}$. Such lattice quantum fields satisfy $\hat{\eta}_{\mathbf{x},\mu}^2 \neq 0$ and $\hat{\eta}_{\mathbf{x},\mu}^3 = 0$. They mutually anticommute at different spatial positions, i.e., $\left\{\hat{\eta}_{\mathbf{x},\mu}, \hat{\eta}_{\mathbf{x}',\mu'}^{(\dagger)}\right\} = 0$ for $\mathbf{x} \neq \mathbf{x}'$ or $\mu \neq \mu'$, and also anticommute with the staggered matter fermionic fields $\left\{\hat{\eta}_{\mathbf{x},\mu}, \hat{\psi}_{\mathbf{x}'}^{(\dagger)}\right\} = 0$[17,18]. Then, we express the comparator on the link as $\hat{U}_{\mathbf{x},\mu} = \hat{\eta}_{\mathbf{x},\mu}\hat{\eta}_{\mathbf{x}+\mu,-\mu}^\dagger$. To explicitly build these three-hardcore fermions for each half-link, we consider two species of standard Dirac fermions $\hat{a}_{\mathbf{x},\mu}$ and $\hat{b}_{\mathbf{x},\mu}$ and we use the following relation:

$$\hat{\eta}_{\mathbf{x},\mu}^\dagger = \hat{n}_{\mathbf{x},\mu}^a \hat{b}_{\mathbf{x},\mu}^\dagger + (1 - \hat{n}_{\mathbf{x},\mu}^b)\hat{a}_{\mathbf{x},\mu}^\dagger \tag{9}$$

where $\hat{n}_{\mathbf{x},\mu}^a$ and $\hat{n}_{\mathbf{x},\mu}^b$ are the occupation number operators for each species, i.e., $\hat{n}_{\mathbf{x},\mu}^a = \hat{a}_{\mathbf{x},\mu}^\dagger \hat{a}_{\mathbf{x},\mu}$ and the same for $\hat{n}_{\mathbf{x},\mu}^b$. For each three-hardcore mode, these operators act on a three-dimensional local Hilbert space with basis $|0\rangle_{\mathbf{x},\mu}$, $|1\rangle_{\mathbf{x},\mu} = \hat{a}_{\mathbf{x},\mu}^\dagger|0\rangle_{\mathbf{x},\mu}$, $|2\rangle_{\mathbf{x},\mu} = \hat{b}_{\mathbf{x},\mu}^\dagger\hat{a}_{\mathbf{x},\mu}^\dagger|0\rangle_{\mathbf{x},\mu}$. In fact, due to the definition in Eq. (9), the algebra of the operators $\hat{\eta}_{\mathbf{x},\mu}$ never accesses the fourth state obtained as $\hat{b}_{\mathbf{x},\mu}^\dagger|0\rangle_{\mathbf{x},\mu}$. By using the same representation on the other half-link through the Dirac operators $\hat{a}_{\mathbf{x}+\mu,-\mu}^\dagger$ and $\hat{b}_{\mathbf{x}+\mu,-\mu}^\dagger$, we would obtain for the complete link a local space of dimension 9. However, the operator that counts the total number of

fermions on the complete link as

$$\hat{L}_{\mathbf{x},\mu} = \hat{n}_{\mathbf{x},\mu}^a + \hat{n}_{\mathbf{x},\mu}^b + \hat{n}_{\mathbf{x}+\mu,-\mu}^a + \hat{n}_{\mathbf{x}+\mu,-\mu}^b, \tag{10}$$

defines asymmetry of the Hamiltonian since it commutes with the operators $\hat{E}_{\mathbf{x},\mu}$ and $\hat{U}_{\mathbf{x},\mu}$. Thus, we can select the sector with $\hat{L}_{\mathbf{x},\mu} = 2$ (two rishons on each full link), reducing the link space to dimension 3 with the basis

$$|\rightarrow\rangle = -|0, 2\rangle = \hat{a}_{\mathbf{x}+\mu,-\mu}^\dagger \hat{b}_{\mathbf{x}+\mu,-\mu}^\dagger|0\rangle_{\mathbf{x},\mu}|0\rangle_{\mathbf{x}+\mu,-\mu},$$
$$|\phi\rangle = |1, 1\rangle = \hat{a}_{\mathbf{x},\mu}^\dagger \hat{a}_{\mathbf{x}+\mu,-\mu}^\dagger|0\rangle_{\mathbf{x},\mu}|0\rangle_{\mathbf{x}+\mu,-\mu}, \tag{11}$$
$$|\leftarrow\rangle = |2, 0\rangle = \hat{b}_{\mathbf{x},\mu}^\dagger \hat{a}_{\mathbf{x},\mu}^\dagger|0\rangle_{\mathbf{x},\mu}|0\rangle_{\mathbf{x}+\mu,-\mu},$$

where the minus sign in the first element allows the operator $\hat{U}_{\mathbf{x},\mu}$ to act correctly following the properties of Eq. (8). By using this representation, the electric field finally corresponds to the imbalance of Dirac fermions between the two halves of the link, so that

$$\hat{E}_{\mathbf{x},\mu} = \frac{1}{2}\left(\hat{n}_{\mathbf{x}+\mu,-\mu}^a + \hat{n}_{\mathbf{x}+\mu,-\mu}^b - \hat{n}_{\mathbf{x},\mu}^a - \hat{n}_{\mathbf{x},\mu}^b\right). \tag{12}$$

This construction in terms of 3-hardcore fermions allows us to define, for each lattice site, a local basis that directly incorporates Gauss's law, by constraining in this way the dynamics to the physical states only. This is a crucial point for both numerical and quantum simulations since non-physical states determine an exponential increase in the complexity of the problem.

From the definition of the link basis states of Eq. (11), it follows that, within the sector with the link-symmetry constraint $\hat{L}_{\mathbf{x},\mu} = 2$, the electric field operator is uniquely identified by taking only the half-link fermionic configuration, namely

$$\hat{E}_{\mathbf{x},\mu} = 1 - \hat{n}_{\mathbf{x},\mu}^a - \hat{n}_{\mathbf{x},\mu}^b. \tag{13}$$

In this way, the generators of the Gauss's law of Eq. (3) are transformed into completely local operators acting on the site $\mathbf{x}$ only

$$\hat{G}_{\mathbf{x}} = \hat{\psi}_{\mathbf{x}}^\dagger \hat{\psi}_{\mathbf{x}} - \frac{1 - (-1)^{\mathbf{x}}}{2} - \sum_\mu\left(1 - \hat{n}_{\mathbf{x},\mu}^a - \hat{n}_{\mathbf{x},\mu}^b\right). \tag{14}$$

Taking into account this property, it is possible to construct the gauge-invariant basis for the local site $\mathbf{x}$, which is composed by the lattice site and the six half-links along with the directions $\pm\mu_x, \pm\mu_y, \pm\mu_z$ (see Fig. 6)

$$\left|\begin{matrix} & & k_6 \\ & k_5 & \\ k_1 & \phi & k_4 \\ & k_2 & \\ & & k_3 \end{matrix}\right\rangle = (-1)^{\delta_{k_1,2}+\delta_{k_2,2}+\delta_{k_3,2}}|\phi\rangle_{\mathbf{x}}$$
$$\times |k_1\rangle_{\mathbf{x},-\mu_x}|k_2\rangle_{\mathbf{x},-\mu_y}|k_3\rangle_{\mathbf{x},-\mu_z} \tag{15}$$
$$\times |k_4\rangle_{\mathbf{x},\mu_x}|k_5\rangle_{\mathbf{x},\mu_y}|k_6\rangle_{\mathbf{x},\mu_z}$$

where $|\phi\rangle_{\mathbf{x}} = (\hat{\psi}_{\mathbf{x}}^\dagger)^\phi|0\rangle$ with $\phi = 0, 1$ describes the presence or the absence of the matter/antimatter particles. The indices $k_j$ run over $\{0, 1, 2\}$ selecting a configuration of the 3-hardcore modes for each respective half-link. The presence

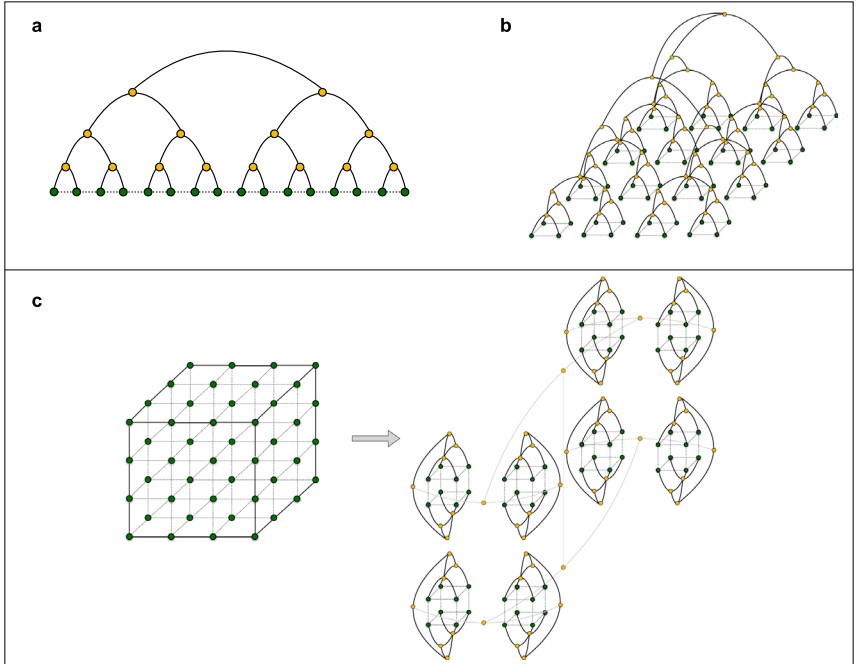

**Fig. 7 TTN ansazts.** TTN representations for **a** 1D lattice and **b** 2D square lattice. Green circles indicate the sites of the lattice connected to the physical indices of the tree, whereas the yellow circles are the tensors making up the TTN. In **c**, we showed our generalization to the 3D cubic lattice that we use for the numerical simulations of the LGT. The different colors of the bond indices are just for better visualization of the tree structure.

of the factor $(-1)^{\delta_{k_1,2}+\delta_{k_2,2}+\delta_{k_3,2}}$ allows us to satisfy the anticommutation relations of the fermionic representation recovering the correct signs of Eq. (11). The occupation numbers $\phi$ and $k_j$ are not independent due to the constraint imposed by Gauss's law

$$\hat{G}_{\mathbf{x}} \left| \begin{matrix} & k_5 & \\ k_1 & \phi & k_4 \\ & k_2 & \end{matrix} \right\rangle^{k_6}_{k_3} = 0. \tag{16}$$

This equation, in the new language of matter fermions and rishons, reads

$$\phi + \sum_{j=1}^{6} k_j = 6 + \frac{1-(-1)^{\mathbf{x}}}{2}. \tag{17}$$

where the factor 6 is indeed the coordination number of the cubic lattice. Thus, the gauge-invariant configurations of the local basis are obtained by applying this constraint, effectively reducing the "dressed-site" (matter and 6 rishon modes) dimension from $2 \times 3^6 = 1458$ to merely 267. We encode these states as building blocks of our computational representation for the TN algorithms. In Fig. 6, we show some examples of gauge-invariant configurations for even and odd sites.

The construction of the gauge-invariant local sites is particularly advantageous for our numerical purposes: in fact, it is now possible to express all the terms in the Hamiltonian of Eqs. (1a)–(1c) of the main text as the product of completely local operators that commute on different sites. Let us consider the kinetic term of the Hamiltonian and apply the representation of the gauge field in terms of the 3-hardcore fermionic modes

$$\hat{\psi}_{\mathbf{x}}^{\dagger} \hat{U}_{\mathbf{x},\mu} \hat{\psi}_{\mathbf{x}+\mu} = \hat{\psi}_{\mathbf{x}}^{\dagger} \hat{\eta}_{\mathbf{x},\mu} \hat{\eta}_{\mathbf{x}+\mu,-\mu}^{\dagger} \hat{\psi}_{\mathbf{x}+\mu}$$
$$= \left( \hat{\eta}_{\mathbf{x},\mu}^{\dagger} \hat{\psi}_{\mathbf{x}} \right)^{\dagger} \left( \hat{\eta}_{\mathbf{x}+\mu,-\mu}^{\dagger} \hat{\psi}_{\mathbf{x}+\mu} \right)$$
$$= M_{\mathbf{x}}^{(\alpha)\dagger} M_{\mathbf{x}+\mu}^{\alpha'} \tag{18}$$

where the indices $\alpha$ and $\alpha'$ select the right operators depending on the different directions in which the hopping process takes place. The operators $M_{\mathbf{x},\mu}^{\alpha}$ are genuinely local (i.e., they commute with operators acting elsewhere) as they are always quadratic in the fermionic operators ($\psi$ and/or $\eta$). The same argument

applies to the magnetic (plaquette) terms in the Hamiltonian

$$\hat{\square}_{\mu_x,\mu_y} = \hat{U}_{\mathbf{x},\mu_x} \hat{U}_{\mathbf{x}+\mu_x,\mu_y} \hat{U}_{\mathbf{x}+\mu_y,\mu_x}^{\dagger} \hat{U}_{\mathbf{x},\mu_y}^{\dagger} =$$
$$= \hat{\eta}_{\mathbf{x},\mu_x} \hat{\eta}_{\mathbf{x}+\mu_x,-\mu_x}^{\dagger} \hat{\eta}_{\mathbf{x}+\mu_x,\mu_y} \hat{\eta}_{\mathbf{x}+\mu_x+\mu_y,-\mu_y}^{\dagger}$$
$$\times \left( \hat{\eta}_{\mathbf{x}+\mu_y,\mu_x} \hat{\eta}_{\mathbf{x}+\mu_x+\mu_y,-\mu_x}^{\dagger} \right)^{\dagger} \left( \hat{\eta}_{\mathbf{x},\mu_y} \hat{\eta}_{\mathbf{x}+\mu_y,-\mu_y}^{\dagger} \right)^{\dagger}$$
$$= -\left( \hat{\eta}_{\mathbf{x},\mu_y}^{\dagger} \hat{\eta}_{\mathbf{x},\mu_x} \right) \left( \hat{\eta}_{\mathbf{x}+\mu_x,-\mu_x}^{\dagger} \hat{\eta}_{\mathbf{x}+\mu_x,\mu_y} \right) \tag{19}$$
$$\times \left( \hat{\eta}_{\mathbf{x}+\mu_x+\mu_y,-\mu_y}^{\dagger} \hat{\eta}_{\mathbf{x}+\mu_x+\mu_y,-\mu_x} \right) \left( \hat{\eta}_{\mathbf{x}+\mu_y,\mu_x}^{\dagger} \hat{\eta}_{\mathbf{x}+\mu_y,-\mu_y} \right)$$
$$\equiv -C_{\mathbf{x}}^{(\alpha)} C_{\mathbf{x}+\mu_x}^{(\alpha')} C_{\mathbf{x}+\mu_x+\mu_y}^{(\alpha'')} C_{\mathbf{x}+\mu_y}^{(\alpha''')},$$

where the indices $\alpha$, $\alpha'$, $\alpha''$, $\alpha'''$ depend on the plane of the plaquette (in this case $x$–$y$) and the links involved in the loop. The operators $C_{\mathbf{x}}^{\alpha}$ are genuinely local and act on the four sites at the corners of the plaquette. The decomposition is the same for the other plaquettes in the planes $x$–$z$ and $y$–$z$. The present construction ensures that they can be treated as spin (or bosonic) operators[97,98], so we can exploit standard TN algorithms, without the need of explicitly implementing the fermionic parity at each site[102–104].

The mass term and the electric field energy in the Hamiltonian of Eqs. (1a)–(1c) of the main text are diagonal in the gauge-invariant basis with the rishon representation and so it is trivial to express them as local operators. These operators include the local chemical potential terms, which we use to pin charges in order to study confinement properties[105–107]. In conclusion, all the operators we employ in the TTN algorithms (see the "Methods" subsection "Tensor Networks") are genuinely local. In order to get an idea of the numerical complexity, we emphasize that the dimension of these matrices acting on the local gauge-invariant basis is $267 \times 267$.

**Tensor networks.** In this section, we present the main concepts of TNs with a particular focus on the TTN ansatz that we exploit in this work[108]. For a detailed and exhaustive description of the subject, please see the technical reviews and textbooks[21,109,110].

Let us consider a generic quantum system composed by $N$ lattice sites, each of which described by a local Hilbert space $H_k$ of finite dimension $d$ and equipped with a local basis $\{|i>_k\}_{1 \le i \le d}$. The whole Hilbert space of the system will be generated by the tensor product of the local Hilbert spaces, that is, $\mathcal{H} = \mathcal{H}_1 \otimes \mathcal{H}_2 \otimes \cdots \mathcal{H}_N$, with a resulting dimension equal to $d^N$. Thus, a generic pure quantum state of the system $|\psi\rangle$ can be expressed as a linear combination of the basis elements of $\mathcal{H}$, i.e.,

$$|\psi\rangle = \sum_{i_1,\dots,i_N=1}^{d} c_{i_1,\dots,i_N} |i_1\rangle_1 \otimes |i_2\rangle_2 \otimes \dots \otimes |i_N\rangle_N. \tag{20}$$

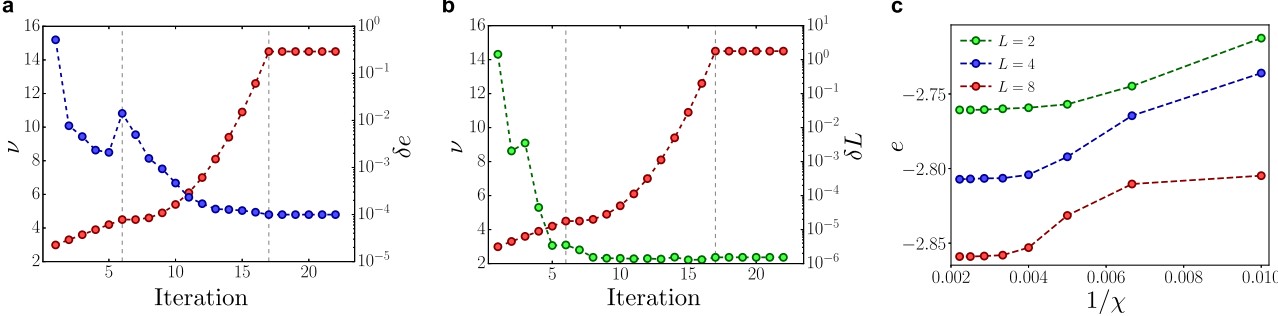

**Fig. 8 Numerical convergence. a** Driven optimization (in three steps: linear, quadratic, constant) of the penalty coefficient $\nu$ (red) and behavior of the energy (blue) as a function of the iterations for an exemplifying simulation. The energy is reported as the difference with the lowest final energy that we reach. **b** Driven optimization of the penalty coefficient $\nu$ (red) and global error $\delta L$ (green) with respect to the link symmetry during the optimization steps. **c** Scaling of the energy density as a function of the inverse of the bond dimension $1/\chi$. The bond dimension $\chi$ is in the range [100, 450].

In principle, the coefficients $c_{i_1,\dots,i_N}$ are $d^N$ complex numbers. As a consequence, this exact representation of the quantum state is completely inefficient from a computational point of view, since it scales exponentially with the system size $N$. In other words, the amount of information that we would need to store in memory for a computational representation of the generic quantum state of the system is exponentially large in the number of degrees of freedom.

However, if we are concerned with local Hamiltonians, which means that a lattice site interacts only with a finite set of neighboring sites and not with all sites of the lattice, it is possible to exploit rigorous results on the scaling of entanglement under a bipartition (area law)[111,112] in order to obtain an efficient representation of the states in the low-energy sectors of such Hamiltonians, e.g., ground-states and first excited states. TN provide a natural language for this representation[113,114] by decomposing the complete rank-$N$ tensor $c_{i_1,\dots,i_N}$ in Eq. (20) into a network of smaller-rank local tensors interconnected with auxiliary indices (bond indices). If we control the dimension of the bond indices with a parameter $\chi$, called the bond dimension, the number of coefficients in the TN is of the order $O(\mathrm{poly}(N)\mathrm{poly}(\chi))$, allowing an efficient representation of the information encoded in the quantum state. Furthermore, the bond dimension $\chi$ is a quantitative estimate of the number of quantum correlations and entanglement present in the TN. In fact, by varying $\chi$, TNs interpolate between a product state ($\chi = 1$) and the exact, inefficient, representation of the considered quantum state ($\chi \approx d^N$).

Matrix product states (MPS) for 1D systems[115–117], projected entangled pair state (PEPS) for 2D and 3D systems[114,118,119], multiscale entanglement renormalization ansatz[120,121] and TTN, that can be defined in any dimension,[108,122,123] are all important examples of efficient representations based on TNs.

MPS algorithms, such as the density matrix renormalization group[124], represent the state-of-the-art technique for the numerical simulation of many-body systems in 1D. MPS satisfy area law and are extremely powerful since they allow to compute scalar products between two wave functions and local observables in an exact and efficient way. This property does not hold true for higher-dimensional generalizations, such as PEPS, and the development of TN algorithms, for accurate and efficiently scalable computations, is at the center of current research efforts.

In particular, one of the main problems is related to the choice of the TN geometry for simulating higher-dimensional systems. PEPS intuitively reproduces the structure of the lattice with one tensor for each physical site and the bond indices directly follow the lattice grid. The resulting TN follows the area-law of entanglement but it contains loops, making the contractions for computing expectation values exponentially hard[125]. Furthermore, the computational cost for performing the variational optimization of PEPS, as for instance in the ground state searching, scales as $O(\chi^{10})$ as a function of the bond dimension. This severely limits the possibility of reaching high values of $\chi$, especially for large system sizes (typical values are $\chi \approx 10$ for spin systems). For our purpose of simulating LGT in three-spatial dimensions, this represents a crucial problem since the local dimension of our model is extremely high, i.e., $d = 267$, and so it becomes necessary to be able to handle high values of $\chi$ in order to reach the numerical convergence.

Alternative ansätze for simulating quantum many-body systems are the TTNs, which decompose the wave function into a network of tensors without loops, allowing efficient contraction algorithms with a polynomial scaling as a function of the system size. In Fig. 7, we show the typical TTN ansazts for 1D and 2D systems and our generalization to the 3D lattice. TTNs offer more tractable computational costs since the complete contraction and the variational optimization algorithms scale as $O(\chi^4)$, making it easier to reach high values of the bond dimension (up to $\chi \approx 1000$). The price to pay for using the loopless structure is related to the area law that TTNs may not explicitly reproduce in dimensions higher than one[126]. Nevertheless, we use the TTN ansatz in a variational optimization, so we can improve the precision by using increasing values of $\chi$, providing in this way a careful control over the convergence of our numerical results.

Ground state computation of our LGT model employs the TTN algorithm for variational ground state search, including the exploitation of Abelian symmetries and the Krylov subspace expansion[110]. The algorithm is implemented to conserve the total charge through the definition of global $U(1)$ symmetry sectors encoded in the TTN. Thus, we can easily access finite charge–density regimes, with an arbitrary imbalance between charges and anticharges.

Our TTN for the 3D lattice is composed entirely of tensors with three links (this structure is usually called *binary tree*). The construction of the TTN starts from merging the physical indices at the bottom, which represent two neighboring lattice sites along the $x$-direction, into one tensor. Then, these tensors are connected along the $y$-direction through new tensors in an upper layer. The tensors in this layer are then connected along the $z$-direction through a new layer of tensors. Thus, this procedure is iteratively repeated by properly setting the connections along with the three spatial directions in the upper layers of the tree. At the beginning of the simulation, we randomly initialize all the tensors in the network and the distribution of the global symmetry sectors. During the variational optimization stage, in order to improve the convergence, we perform the single-tensor optimization with subspace-expansion technique, i.e., allowing a dynamical increase of the local bond dimension and adapting the symmetry sectors[110]. This scheme has a global computational cost of the order $O(\chi^4)$. The single tensor optimization is implemented in three steps: (i) the effective Hamiltonian $H_{\mathrm{eff}}$ for the tensor is obtained by contracting the complete Hamiltonian of the system with all the remaining tensors of the tree; (ii) the local eigenvalue problem for $H_{\mathrm{eff}}$ is solved by using the Arnoldi method of the ARPACK library; (iii) the tensor is updated by the eigenvector of $H_{\mathrm{eff}}$ corresponding to the lowest eigenvalue. This procedure is iterated by sweeping through the TTN from the lowest to the highest layers, gradually reducing the energy expectation value. After completing the whole sweep, the procedure is iterated again and again, until the desired convergence in the energy is reached. The precision of the Arnoldi algorithm is increased in each sweep, for gaining more accuracy in solving the local eigenvalue problems as we approach the final convergence.

TTN computations presented in this work are extremely challenging due to the complexity of LGTs in the three-dimensional scenario. They were performed on different HPC-clusters (CloudVeneto, CINECA, BwUniCluster, and ATOS Bull): a single simulation for the maximum size that we reached, an $8 \times 8 \times 8$ lattice, can last up to five weeks until final convergence, depending on the different regimes of the model and the control parameters of the algorithms.

**Numerical convergence**. With our numerical simulations, we characterize the properties of the ground state of the system as a function of the parameters in the Hamiltonian of Eqs. (1a)–(1c) of the main text. We fix the energy scale by setting the hopping coefficient $t = 1$ and we access several regimes of the mass $m$, the electric $g_{\mathrm{e}}$ and the magnetic coupling $g_{\mathrm{m}}$. We consider simple cubic lattices $L \times L \times L$ with the linear size $L$ being a binary power; in particular, we simulate the case with $L = 2, 4, 8$, that is, up to 512 lattice sites.

As explained in the "Methods" subsection "Fermionic compact representation of local gauge-invariant site", in order to obtain the right representation of the electric field operators, we have to enforce the extra link symmetry constraint $\hat{L}_{\mathbf{x},\mu} = 2$ at every pair of neighboring sites. For this reason, we include in the Hamiltonian additional terms that energetically penalize all the states with a number of hardcore fermions per link different from two, namely

$$\hat{H}_{\mathrm{pen}} = \nu \sum_{\mathbf{x},\mu} \left(1 - \hat{\delta}_{2,\hat{L}_{\mathbf{x},\mu}}\right) \qquad (21)$$

where $\nu > 0$ is the penalty coefficient and $\hat{\delta}_{2,\hat{L}_{\mathbf{x},\mu}}$ are the projectors on the states that satisfy the extra link constraint. In this way, the penalty terms vanish when the link symmetry is satisfied and raise the energy of the states violating the constraint. In principle, the link symmetry is rigorously satisfied for $\nu \to \infty$. At the numerical

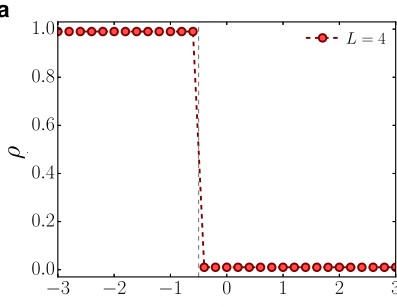
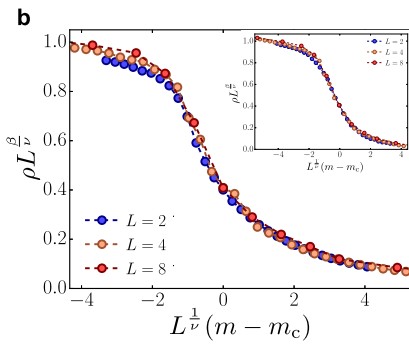
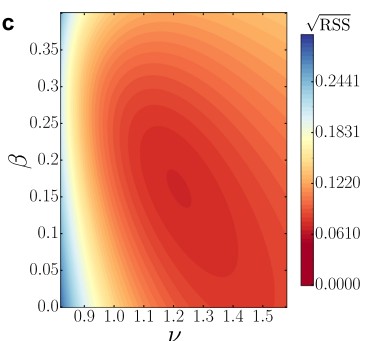

**Fig. 9 Finite-size scaling analysis. a** Particle density as a function of $m$, for $t = 0$, $g_m^2 = 0$ and $L = 4$. **b** Universal scaling function $\lambda(x)$ close to the transition point $m_c \approx -0.39$ for $g_m^2 = 0$ with critical exponents $\beta \approx 0.16$ and $\nu \approx 1.22$. The inset shows the same universal behavior close to the transition point $m_c \approx 0.22$ in the presence of magnetic interactions with $g_m^2 = 8/g_e^2 = 4$ and the same critical exponents $\beta \approx 0.16$ and $\nu \approx 1.22$. **c** Contour plot of the square root of the residual sum of squares in the $(\nu, \beta)$ plane for the best-fitting values of the critical exponents.

level, this limit translates into choosing $\nu$ much larger than the other simulation parameters of the Hamiltonian, i.e., $\nu \gg \max\{|t|, |m|, |g_e|, |g_m|\}$. However, setting $\nu$ too large in the first optimization steps could lead to local minima or non-physical states, since the variational algorithm would focus only on the penalty terms more than the physical ones. In order to avoid this problem and reach the convergence, we adopt a driven optimization, by varying the penalty coefficient $\nu$ in three steps: (i) starting from a very small value of $\nu$ and from a random state of the TTN, that in general does not respect the extra link symmetry, we drive the penalty term with linear growth of $\nu$ during the first optimization sweeps. In this stage, the optimization will focus mainly on the physical quantities, until we notice a slight rise in the energy: this effect signals that the global optimization procedure of the TTN becomes significantly sensitive to the penalty terms. (ii) Thus, we impose a quadratic growth of $\nu$ so that, in the immediately following sweeps, the penalty is increased at a slower rate with respect to the linear regime. (iii) After reaching the maximum desired value of $\nu$, which is an input parameter of the simulation, we keep it fixed, performing the last sweeps in order to ensure the convergence of the energy. This driven optimization strategy is summarized in Fig. 8a where we show the three different stages of the penalty coefficient $\nu$ and typical behavior of the energy difference $\delta e$, computed with respect to the lowest final energy that we reach, as a function of the iterations.

We can also quantify the global error with respect to the link symmetry during the driven optimization sweeps, by defining

$$\delta L = \sum_{\mathbf{x}, \mu} \left| \langle GS | (\hat{L}_{\mathbf{x},\mu} - 2) | GS \rangle \right| \tag{22}$$

i.e., the sum of the deviations from the exact link constraint $\hat{L}_{\mathbf{x},\mu} = 2$, computed over all the links of the lattice on the ground state. The typical behavior of this quantity is shown in Fig. 8b: at the end of the optimization procedure, the global error results of the order of $10^{-6}$. We also check the convergence of our TTN algorithms as a function of the bond dimension $\chi$, by using $\chi = 450$ at most to ensure the stability of our findings. Depending on the different system sizes and regimes of physical parameters, we estimate the relative error of the energy in the range $[10^{-2}, 10^{-4}]$. A typical scaling of the energy density as a function of the inverse of the bond dimension $1/\chi$ is shown in Fig. 8c.

**Critical points: scaling analysis.** In this section, we show the finite-size scaling analysis for detecting the phase transition separating the charge-crystal phase and the vacuum phase and the related location of the critical points.

At $t = 0$ and neglecting the magnetic interactions, i.e., for $g_m^2 = 0$, the Hamiltonian of Eq. (1a)–(1c) results diagonal in the local basis described in the "Methods" subsection "Fermionic compact representation of local gauge-invariant site" and it is trivial to prove that the system undergoes a first-order phase transition between the bare vacuum, with energy $E_v = -m\frac{L^3}{2}$ and the charge-crystal phase, with energy $E_{ch} = (m + \frac{g_e^2}{2})\frac{L^3}{2}$. The ground-state exhibits a level-crossing at the critical value $m_c^{(0)} = -\frac{g_e^2}{4} = -\frac{1}{2}$ that is obtained at $E_v = E_{ch}$. This behavior is clearly seen in Fig. 9a, showing a discontinuous transition between the two configurations.

In order to understand the behavior of the system for finite $t = 1$ and $g_m^2 = 0$, we observe that the density, plotted in Fig. 2a of the main text, changes continuously as a function of the mass parameter and we might have a second-order phase transition. Finite-size scaling theory[127] implies that the behavior of the system close to a critical point, i.e., for $m \approx m_c$, can be described in terms of a

universal function $\lambda(x)$ such that for our observable

$$\rho L^{\frac{\beta}{\nu}} = \lambda \left( L^{\frac{1}{\nu}} (m - m_c) \right) \tag{23}$$

where $\beta$ and $\nu$ are critical exponents. In particular, this relation implies that for $m \approx m_c$, the value of $\rho L^{\frac{\beta}{\nu}}$ is independent of the size of the system. We use this property to get an estimate of the values of $m_c, \beta, \nu$. In particular, we consider a grid of values for this parameter, and for each set of values, we fit our points $\rho L^{\frac{\beta}{\nu}}$ with an high-degree polynomial $f\left( L^{\frac{1}{\nu}} (m - m_c) \right)$. We compute the residual sum of squares (RSS) and we select the set of values which minimize this quantity, producing the best data collapse. We get for the critical point $m_c \approx -0.39$ and for the critical exponents $\beta \approx 0.16$ and $\nu \approx 1.22$. In Fig. 9(b) we show the collapse of our numerical results onto the same universal function $\lambda(x)$ and in Fig. 9(c) a contour plot of the square root of the RSS in the $(\nu, \beta)$ plane for the best-fitting values.

By extending the previous considerations and the finite-size scaling analysis to the case with magnetic interactions with $g_m^2 = 8/g_e^2 = 4$, we check again the presence of a critical point and the values of critical exponents through the formula of Eq. (23). We obtain a universal scaling function for $m_c \approx 0.22$ and the same critical exponents $\beta \approx 0.16$, $\nu \approx 1.22$, as reported in the inset of Fig. 9b. Thus, while the transition and its universality remain unchanged in the presence of the magnetic coupling, the critical point is shifted toward positive values of the mass parameter, signaling that the magnetic interactions determine a visible enhancement of the production of charges and anti charges out of the vacuum.

Although a more precise determination of the numerical values of the critical exponents would require additional extensive analysis that results beyond the scope of this paper, our findings strongly indicate the presence of a phase transition at finite $m$ for the three-dimensional lattice model of QED (compare with other previously investigated transitions in lattice QED, e.g., ref. [128]).

## Data availability

The data that support the findings of this study are available from the corresponding author upon reasonable request. Source data of the figures are attached to the manuscript. Source data are provided with this paper.

## Code availability

The authors are available for discussing details of the implementation of the computer codes developed and used in this study upon reasonable request.

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

## Acknowledgements

Authors kindly acknowledge support from the Italian PRIN2017 and Fondazione CARIPARO, the Horizon 2020 research and innovation program under grant agreement No. 817482 (Quantum Flagship—PASQuanS), the QuantERA projects QTFLAG and QuantHEP, the DFG project TWITTER, the INFN project QUANTUM, and the Austrian Research Promotion Agency (FFG) via QFTE project AutomatiQ. We acknowledge computational resources by the Cloud Veneto, CINECA, the BwUniCluster, and ATOS Bull.

## Author contributions

G.M. and T.F. implemented the TTN algorithms for the three-dimensional model, basing on a code previously developed by T.F. for 2D LGT; G.M. performed the numerical simulations and analyzed the results; P.S. provided the basis for the theoretical strategy of mapping the lattice gauge theory into the proper computational framework. The interpretation of the results was mainly done by G.M., P.S., and S.M.; G.M. and P.S. wrote the paper; S.M. conceived, supervised, and directed the project. All authors discussed the results, contributed to refining the paper, and approved it.

## Competing interests

The authors declare no competing interests.
