## [Peer Review File · Nature Communications]

Reviewers' Comments:

Reviewer #1:

Remarks to the Author:

Tensor network (TN) techniques have turned out to be very useful in many branches of computational physics, including lattice gauge theories. One of the crucial motivations is to be able to investigate theories where the notorious sign problem in the Monte Carlo approach prevents access to important parameter regimes, e.g. QCD at finite density. Hence, if TN methods can indeed tackle QCD, one can expect a vital breakthrough in our understanding of the strong interaction from first principles. As the authors point out, TN achieved remarkable successes for lattice gauge theories in 1+1 dimensions and have shown potential also in 2+1 dimensions. However, the real-world case of 3+1 dimensions has not been addressed so far and it is crucial to show the perspectives of TN also here, starting obviously with a theory simpler than QCD. Importantly, it is essential not only to tackle a small 3+1-dimensional system, but to be able to demonstrate that the setup has enough potential to be extended to larger system sizes. This boils down to proper scaling of entanglement, i.e. that the TN Ansatz obeys the area law, which in practice makes the required bond dimension independent of the size or at worst having only logarithmic corrections.

The authors use the TN Ansatz of unconstrained tree tensor network (TTN) to handle compact QED in three spatial dimensions, truncated to the lowest non-trivial spin representation in the quantum link formalism. While the theory "sounds" akin to real-world QED, its properties are pretty much different and it is used as a toy model for charge confinement. In a way, being different from real-world QED lessens the interest of the phenomenological particle physics community who might want to see a demonstration of TN application to an actual real-world theory. However, compact QED is a very natural formulation for the lattice and thus, a good starting point for demonstrating the potential of TN in 3+1 dimensions.

The paper is very well-written and contains a selection of interesting results, being able to convincingly demonstrate the usefulness of TTN for small systems in 3 spatial dimensions. The main part of the article concentrates on general discussion of the motivation and the model and presents the quite robust physical conclusions that could be drawn from the simulations. The technical details are given in the appendices, together with a broad discussion on the TN approach in general and TTN in particular.

I am missing one crucial element that is essential to draw conclusions about the potential of the approach in the long run. This would be a demonstration that larger system sizes are also possible to achieve with TTN at reasonable bond dimensions. However, I am afraid such a demonstration is not provided for the present work, because it is well-known that TTN "fail to capture area law for large system sizes". The latter is a quotation from a paper by some of the present authors, 2011.08200, where they propose an extension of TTN to improve this aspect. Thus, they are certainly aware of the limitations of the simple TTN approach of the present paper. I find these limitations hard to reconcile with the general message a TN non-expert reader could draw from the paper - that this is an important milestone for TN on the way for simulating quantum field theories in 3+1 dimensions, in particular QCD one day. I would rather draw a more modest conclusion that even relatively simple TN Ansatzes can do well if the system size is small enough. However, this proceeds not via making use of the beautiful quantum information basis behind TN, i.e. obeying area laws, but rather by brute-force increase of the bond dimension to the level just enough to describe the physics at a given system size. In this sense, this is similar to brute-force applications of DMRG for systems in more than one spatial dimension. It is true that some of the most precise results for many 2D quantum many-body systems still come from DMRG extended to huge bond dimensions and this is in many cases good enough to draw genuine physical conclusions. Nevertheless, TN for real-world particle physics can be useful only if system sizes significantly larger than in the present paper are reached and I do not find a convincing demonstration that simple TTN can get close to this aim.

The above significantly lowers my assessment of the fulfillment of Nature Communications publication criteria. I believe this is very interesting and well-reported work, but its importance is limited in my view. As such, I think the paper should be published in a more specialized journal,

upon inserting a discussion of the limitations of simple TTN and possible better ways, whether it be PEPS or a smaller family of TN Ansatzes, such as augmented TTN advocated by some of the present authors. Another, more minor reservation that I have is that it is not clear to me how practical it is to include higher-spin representations. It is clear this is theoretically possible, but I consider it a very worthwhile direction of work to include these in practice, e.g. as extension of the previous work of some of the present authors for 2+1-dimensional QED. Actually, it is very important to demonstrate that the investigated model can indeed be called QED. This could proceed by including the next non-trivial representation(s) and showing that its/their effect gets saturated. Without this, it is a bit of a stretch to use the name QED. However, I consider it to be more minor and I believe that a scalable TN Ansatz for truncated spin-one QED would already be a magnificent achievement worth wide dissemination.

Reviewer #2:

Remarks to the Author:

Tensor networks form a very promising avenue for the study of complex quantum many-body systems in a "classical" way, by focusing only on the physically relevant set of states within the exponentially large Hilbert space of such systems. These states are special in their entanglement properties (area law), and the various tensor network constructions guarantee that. Besides, these states are equipped with an impressive numerical toolbox which allows one to go beyond the capabilities of other methods, such as quantum Monte Carlo, avoiding the sign problem in the presence of fermions with a finite chemical potential. Tensor network methods have been extremely useful, showing remarkable results in condensed matter physics, and in the recent years the formalism and numerical toolbox has been extended impressively to particle physics too, especially lattice gauge theories, by several research groups, including the authors of this manuscript.

In their paper, Magnifico, Felser, Silvi and Montangelo present truly remarkable results: a numerical study of a 3+1d truncated $U(1)$ lattice gauge theory, with fermionic matter, using a tree tensor network. It certainly is a pioneering work in the sense that it shows three dimensional numerical results, using tensor networks, for a lattice gauge theory. The authors have mapped the relevant Hamiltonian into terms that may be treated using current tree tensor network machinery, which allowed them to run optimization algorithms and find the ground state of a minimally truncated $U(1)$ Kogut-Susskind Hamiltonian in 3+1d with staggered fermionic matter. The authors have done it for various coupling constants, including finite density scenarios, and managed to see both of the expected phases: confinement (linear potential) for large coupling and Coulomb (for small coupling), and the transition between the two. They have constructed a beautiful demonstration of a quantum capacitor as an equilibrium counterpart of the dynamical Schwinger mechanism, allowing them to study string breaking without dynamics; and also computed the surface charge density - which demonstrates, quantitatively, the right finite density effects.

The authors explain the model they study very clearly in section I, with further details in Appendix A. As explained in Appendix A, they map the fermionic degrees of freedom to bosonic operators, arriving at a bosonic Hamiltonian on which tensor networks techniques could be applied. All these technical details are described within the appendices, leaving the results and the physical discussion to the main part of the paper which I find reasonable. Appendix B introduces tensor networks very well with proper referencing, and C discusses the numerical convergence of the computations carried out. Appendix D is not directly related to TNs but very clearly explains the scaling analysis of the results.

Back to the main text, in sections II, III, IV and V the authors discuss the results - zero temperature phase transition between the two phases, quantum capacitor, confinement properties (including the beautiful figure 4 showing the linear/Coulomb potentials obtained numerically in both phases) and the study of finite density. The properties of the computed observables and manifested physical phenomena are generally well explained in the text, and the figures deliver the results very clearly.

I certainly find this work novel and important and recommend its publication. However, prior to

that there are several issues that, in my opinion, have to be addressed by the authors, and I shall list them hereby.

1. The model. The authors refer to [16] as the source for their Hamiltonian and fermionic formulation. There is no doubt that in this seminal paper, Kogut and Susskind have introduced, for the first time, the Hamiltonian formulation of lattice gauge theories. However, the case studied in that paper was of a non-Abelian lattice gauge theory, where the gauge group was $SU(2)$, and mostly with the pure gauge case, with no dynamical fermions.

I believe that while reference [16] should be kept, it would be useful to refer also the other references where the $U(1)$ Hamiltonian was explicitly introduced, such as Kogut's review, *Rev. Mod. Phys.* 51, 659 from 1979.

Regarding the fermions, in [16] they were only briefly discussed, and the staggering was done in another, earlier form, addressing the doubling problem only partially. There, on each site they placed a two component spinor, uniting to four component ones in the continuum limit, where one sublattice describes particles and the other anti-particles. The formulation with a single fermionic mode per site, used by the authors for their tensor network computation, which is the later, commonly used now form of staggered fermions, was introduced by Susskind alone, in *Phys. Rev. D* 16, 3031, 1977.

Furthermore, regarding the Gauss laws. The authors neglect all the static charge sectors, besides the one with no static charges. This is a fairly reasonable physical choice which is commonly made. However, when the Gauss law is introduced after eq. (3), when the authors state that the generators G annihilate the physical quantum states, it should be made more clear; in general they are eigenstates of the G operators, and only within the sector with no static charges the eigenvalue is zero. This may also create confusion with the half-filling choice they make in the beginning of section II: the authors write "we focus on the zero charge sector..." however, it may be more accurate to call it the net zero dynamical charge sector.

2. References. I find the referencing of previous works slightly partial, with respect to four different types of previous works:

- The list [21-28] of one dimensional works lacks some important works of several authors, including some of the manuscript's authors as well. The choice of these particular works to be cited here (and several others only later in the context of truncating the Hilbert space in Appendix A - 51, 52, 54, 55, 57 and 59) is not entirely clear to me. There are many other works which could be equally mentioned (e.g. *JHEP* 11 (2013) 158, 10.1007/JHEP07(2015)130, *Phys. Rev. D* 96, 114501 (2017), *Phys. Rev. D* 94, 085018 (2016), *Phys. Rev. X* 6, 041040 (2016), *Phys. Rev. D* 92, 034519 (2015), *Phys. Rev. X* 7, 041046 (2017), *Ann. phys.* 386, 199 (2017) and others). A possible way out to deal with a partial choice of references would be to cite a review, since the current work does not focus on one dimensional models. Furthermore, at least work 23, by the way, represents a rather high dimensional scenario (and not necessarily 1+1 as in the context it appears here).

- As for the works in more than one space dimension, which are more relevant for the manuscript in discussion, the list [29-33] represents very well all the collaborations and groups working in the field (with the addition of cited with the one dimensional works). However, some are missing, such as *Phys. Rev. B* 83, 115127 (2011), or *Phys. Rev. D* 97, 034510 (2018), which introduces the combination of Monte Carlo used later in [33], forming another way that would be feasible in 3+1d (a particular case of it is discussed in reference 33, but it does not include the entire picture). One may also wish to consider the completely parallel line of works on tensor networks for LGTs, tensor field theory, which is summarized in the review paper arXiv:2010.06539 [hep-lat].

- Regarding the quantum simulation/computation works cited. These appear mainly in two different places, mostly related to effects of truncating the local gauge field Hilbert spaces: [36-42] in the beginning, as well as further ones in Appendix A. The focus, mainly in the introduction, is on experimental works, and all these works should be indeed cited. The theory works mentioned in the introduction include a newer [41] and an older one [42] - both important and relevant; however, as the field of quantum simulation has started with several theoretical works, from the early 2010s, where the issue of gauge field truncation has been extensively discussed, mostly by the ICFO, MPQ-TAU, Innsbruck and Heidelberg collaborations (including works dealing in particular

with truncation effects, such as Phys. Rev. A. 88, 023617 (2013), Phys. Rev. D 91, 054506 (2015) and New J. Phys. 19, 023030 (2017)). Such works are not represented (with the exception of one of the Innsbruck works [75] cited later on in the appendix). A parallel approach is to go to a dual formulation, as was done recently in several works, including 64 cited by the authors, but also Phys. Rev. D 102, 094515 (2020) and Phys. Rev. D 102, 114517 (2020).

I would strongly suggest, for the sake of completeness, to include at least the relevant review articles (Contemporary Physics 57 388 (2016), Reports on Progress in Physics 79 (1), 014401 (2015) and Ann. Phys. 525, 777 (2013)). On the other hand some of the cited works do not deal with truncation at all, by either eliminating the gauge field, simulating a Z2 LGT or not dealing with gauge fields at all. No citation should be removed but I encourage the authors to reorganize that parts; perhaps some of the citations should be made instead in the beginning of App. A, where the truncation is visited in further detail.

- There are also other "quantum based" methods which could be used for LGT computations in more than 1+1d. One example is Phys. Rev. Research 2, 043145 (2020), where a generalization of Gaussian states enables to describe real time dynamics of a lattice gauge theory.

3. Very minor issues:

- In equation (A2), is the minus sign before the zero ket at the last equation required?
- Last paragraph of the first column of page 10: LTG should be LGT?

To conclude, I believe that this is a beautiful paper reporting important and ground-breaking work, which should be published, once the above issues are addressed by the authors.

Reviewer #3:

Remarks to the Author:

The authors use a Tree tensor Network to variationally approximate the ground state of compact QED in 3+1 dimension.

To the best of my knowledge, this is the first time that this type of calculation is carried in 3+1 dimensions for a lattice gauge theory at finite density and this is a very interesting new direction of research. I recommend publication provided that the authors address the questions below in a satisfactory manner.

1) The staggered fermions that are used in numerical Lagrangian calculations in 3+1 dimensions have signs that appear in the Dirac operator and discrete symmetries associated with rotations and shifts (see, for instance, Kilcup and Sharpe, Nuclear Physics B283, 493 section 2). I don't see these sign factors in the Hamiltonian of Eq. (1a). Are the authors using a different type of staggered fermions?

2) The numerical calculations can be done for independent (t, m, g_e, g_m) , however the author refer to a "physical regime" where $g_e g_m = 2\sqrt{2}t$. What is the physics? QED as we know it or some condensed matter model, or ...?

3) The authors call g , the QED coupling. The dimension of g is a bit obscure. In Eq. (4), g^2 has the dimension of an energy/distance, in Eq.(5) the dimension of an energy. In continuous QED in 3+1 dimension, the coupling is usually dimensionless. Please clarify.

3) In section III (calculation with the capacitor), the authors introduce large chemical potentials on the plates. In section V (finite density) they "inject" the charges which end up on the boundary. It is not clear how this is accomplished. Would it be possible to use a uniform chemical potential to create a uniform charge density? Such a calculations have been performed in 3+1 compact QED with a large mass (Kniely and Gattringer, arXiv 1502.00788).

4) Does the crystal phase have known physical realizations? Is the universality class of the second order phase transition to this phase known?

5) Is there any way to test the computational methods with independent numerical results? For instance, there are calculations of fermion condensates (without chemical potential) in 3+1 compact QED by Kogut and collaborators (PRL 61, 2416 and later publications).

Minor presentation details:

The last sentence of the abstract says that fundamental questions are addressed. It would be good to say what the findings are.

First paragraph of the introduction: ...(QCD), a founding pillar ...for almost a century. I believe the authors mean "half a century" (or I misunderstood the sentence).

————— **Reply to Reviewer 1:** —————

We warmly thank Referee 1 for stating that “the paper is very well-written and contains a selection of interesting results, being able to convincingly demonstrate the usefulness of TTN for small systems in 3 spatial dimensions” and “the main part of the article [...] presents the quite robust physical conclusions that could be drawn from the simulations”. Referee 1 raised some concerns on the applicability perspective of Tree Tensor Networks (TTN) approach for larger system sizes that would be needed for tackling the continuum limit of the real-world particle physics. The Referee also offered us relevant proposals that could be carried out onto our numerical architecture, such as the study of higher-spin representations of the model. We find these suggestions extremely exciting and, in principle, definitely fitting for TTN methods, with no formal or theoretical limitations. Unfortunately, we can not but consider these investigations a completely novel and stand-alone research project. Indeed, from a purely technical point of view, these analyses are too time-consuming both for user- and computer run-time and go well beyond the scope of the current manuscript. In the following, we provide a comprehensive discussion on the comments of Referee 1, mainly focused on the scalability perspective of our approach.

We hope that the Referee 1 is satisfied with our arguments, in the light that the remaining ideas will surely provide material for future research.

—

Referee: Tensor network (TN) techniques have turned out to be very useful in many branches of computational physics, including lattice gauge theories. One of the crucial motivations is to be able to investigate theories where the notorious sign problem in the Monte Carlo approach prevents access to important parameter regimes, e.g. QCD at finite density. Hence, if TN methods can indeed tackle QCD, one can expect a vital breakthrough in our understanding of the strong interaction from first principles. As the authors point out, TN achieved remarkable successes for lattice gauge theories in 1+1 dimensions and have shown potential also in 2+1 dimensions. However, the real-world case of 3+1 dimensions has not been addressed so far and it is crucial to show the perspectives of TN also here, starting obviously with a theory simpler than QCD. Importantly, it is essential not only to tackle a small 3+1-dimensional system, but to be able to demonstrate that the setup has enough potential to be extended to larger system sizes. This boils down to proper scaling of entanglement, i.e. that the TN Ansatz obeys the area law, which in practice makes the required bond dimension independent of the size or at worst having only logarithmic corrections.

Authors: We agree with these statements, but we would like to point out some important properties related to the entanglement scaling at a general level. Most of the interesting phases of matter, even if generated by complicated Hamiltonian such as the ones of Lattice Gauge Theories, actually show a low or intermediate entanglement content. In particular, in 3 spatial dimensions, it is expected that ground states typically demand a smaller bond dimension than their counterparts in lower dimensions, because the entanglement of a physical site is shared with more neighbor sites, such that low-energy states

typically are closer to product states. In other words, in 3 dimensions, the physics of the ground states should be well-approximated by the (cluster) mean-field approach and, even for very large system sizes, it is expected that the required bond dimension remains moderate and stable. Thus, TN structures such as TTN, that does not strictly encode area law, but at the same time benefits from a favorable numerical scaling with the bond dimension, might be appropriate for very larger sizes. We also stress that the bond dimension that we reach in this work ($\chi \sim 450$) and the limited sizes ensure the convergence of our numerical simulations.

–

Referee: The authors use the TN Ansatz of unconstrained tree tensor network (TTN) to handle compact QED in three spatial dimensions, truncated to the lowest non-trivial spin representation in the quantum link formalism. While the theory “sounds” akin to real-world QED, its properties are pretty much different and it is used as a toy model for charge confinement. In a way, being different from real-world QED lessens the interest of the phenomenological particle physics community who might want to see a demonstration of TN application to an actual real-world theory. However, compact QED is a very natural formulation for the lattice and thus, a good starting point for demonstrating the potential of TN in 3+1 dimensions.

Authors: We completely agree. The interest in compact QED lies in the fact that this model shares several non-perturbative features with much more complicated LGTs, such as QCD, and thus it represents a very natural starting point for testing the capabilities of TN approach in higher-dimensions. Furthermore, the extension to non-abelian symmetries is straightforward in the context of TN, as shown in New J. Phys. 16 103015 (2014) and applied, for one-dimensional non-abelian LGTs, in Quantum 1, 9 (2017) and Phys. Rev. D 100, 074512 (2019)). This will surely provide an interesting and promising avenue for future research with the ultimate goal of studying similar problems, like confinement properties, in the realistic scenario of QCD.

–

Referee: The paper is very well-written and contains a selection of interesting results, being able to convincingly demonstrate the usefulness of TTN for small systems in 3 spatial dimensions. The main part of the article concentrates on general discussion of the motivation and the model and presents the quite robust physical conclusions that could be drawn from the simulations. The technical details are given in the appendices, together with a broad discussion on the TN approach in general and TTN in particular.

I am missing one crucial element that is essential to draw conclusions about the potential of the approach in the long run. This would be a demonstration that larger system sizes are also possible to achieve with TTN at reasonable bond dimensions. However, I am afraid such a demonstration is not provided for the present work, because it is well-known that TTN “fail to capture area law for large system sizes”. The latter is a quotation from a paper by some of the present authors, 2011.08200, where they propose an extension of TTN to improve this aspect. Thus, they are certainly aware of the limitations of the simple TTN

approach of the present paper. I find these limitations hard to reconcile with the general message a TN non-expert reader could draw from the paper - that this is an important milestone for TN on the way for simulating quantum field theories in 3+1 dimensions, in particular QCD one day. I would rather draw a more modest conclusion that even relatively simple TN Ansatz can do well if the system size is small enough. However, this proceeds not via making use of the beautiful quantum information basis behind TN, i.e. obeying area laws, but rather by brute-force increase of the bond dimension to the level just enough to describe the physics at a given system size. In this sense, this is similar to brute-force applications of DMRG for systems in more than one spatial dimension. It is true that some of the most precise results for many 2D quantum many-body systems still come from DMRG extended to huge bond dimensions and this is in many cases good enough to draw genuine physical conclusions. Nevertheless, TN for real-world particle physics can be useful only if system sizes significantly larger than in the present paper are reached and I do not find a convincing demonstration that simple TTN can get close to this aim.

Authors: The Referee is correct. Indeed, the augmented Tree Tensor Networks (aTTN), presented in the paper 2011.08200, can be used to improve the entanglement scaling as already demonstrated in other settings. However, this is just a technically challenging but straightforward development of this work and this is exactly why we believe that it is a potential seminal work. In addition, let us point out that, as detailed in the paper 2011.08200, the improvement in the ground-state energy offered by the aTTN is evident only when the typical lattice sizes considered are well beyond the one reached in the present work, otherwise no difference between TTN and aTTN was highlighted, supporting the validity of our findings of the present paper.

We have of course to scale our algorithms for larger sizes exactly at the same level of past and current Monte Carlo simulations of LTGs. In this regard, we would like to point out that, to the best of our knowledge, the largest lattice simulated to date has a linear size (in the spatial directions) of $L = 144$, obtained with an extreme-scale technical parallelization of Monte Carlo algorithms on thousands of CPU/GPU computational nodes (Phys. Rev. D 98, 030001 (2018)). This value is just an order of magnitude below the maximum size that we reach in our work and we have not exploited yet the power of a massive parallelization on multi-nodes architectures (e.g. MPI library) or the benefits offered by GPU tensor contractions. From this point of view, taking into account the very large dimension of the local basis of the LTG models we consider, the challenge now is not the entanglement content but the parallel computation of the huge number of operators and tensors to be optimised (more than ten thousands!) that, as we observed by profiling the execution of our codes, represents the most time-consuming part of the algorithm. We know how to scale our algorithms by following the principles of High-Performance Computing and we are already working in this direction with some preliminary tests. However, such an investigation is extremely time-consuming compared to the additional insight it provides, and there is no reason it shall not work for increasing lattice sizes. In particular, we expect an almost linear improvement with the number of computational nodes, and thus we are

confident to gain at least one order of magnitude (highly conservative estimate) and another two from GPU parallelization (again conservative), resulting in a massive speedup that will allow to get at least a factor 8 in every spatial dimensions. In conclusion, we are extremely confident that we can scale the system size in the very near future and that all the preliminary work presented in this manuscript forms a strong basis to support our statement.

In the Outlook section, we added a discussion on this important point that was not stressed enough in the previous version of the manuscript. We hope that the Referee now shares our enthusiastic but highly solid view of the next steps.

—

Referee: The above significantly lowers my assessment of the fulfillment of Nature Communications publication criteria. I believe this is very interesting and well-reported work, but its importance is limited in my view. As such, I think the paper should be published in a more specialized journal, upon inserting a discussion of the limitations of simple TTN and possible better ways, whether it be PEPS or a smaller family of TN Ansatz, such as augmented TTN advocated by some of the present authors. Another, more minor reservation that I have is that it is not clear to me how practical it is to include higher-spin representations. It is clear this is theoretically possible, but I consider it a very worthwhile direction of work to include these in practice, e.g. as extension of the previous work of some of the present authors for 2+1-dimensional QED. Actually, it is very important to demonstrate that the investigated model can indeed be called QED. This could proceed by including the next non-trivial representation(s) and showing that its/their effect gets saturated. Without this, it is a bit of a stretch to use the name QED. However, I consider it to be more minor and I believe that a scalable TN Ansatz for truncated spin-one QED would already be a magnificent achievement worth wide dissemination.

Authors: Indeed this is a very important but minor point and given that we are dealing with a local property, all the previous results presented in 1D apply straightforwardly. In Refs. Phys. Rev. D 95, 094509 (2017), Phys. Rev. Lett. 112, 201601 (2014), Phys. Rev. D 98, 074503 (2018), it has been already shown that the higher-spin representation (or similarly Z_n symmetries for large- n) can be easily treated with TN methods and actually that the convergence is pretty fast. Again, we thank the Referee for stressing this important point and we added a comment on this in the Outlook section.

————— Reply to Reviewer 2: —————

We warmly thank Referee 2 for sharing our enthusiasm regarding our contribution and its impact. Referee 2 put forward a few suggestion on how to better clarify some of the results, and especially how to better contextualize our work. We welcomed Referee 2's advice, which lead us to expand and substantially rearrange our extensive bibliography (while still complying to the journal's guidelines).

–
Referee: The model. The authors refer to [16] as the source for their Hamiltonian and fermionic formulation. There is no doubt that in this seminal paper, Kogut and Susskind have introduced, for the first time, the Hamiltonian formulation of lattice gauge theories. However, the case studied in that paper was of a non-Abelian lattice gauge theory, where the gauge group was $SU(2)$, and mostly with the pure gauge case, with no dynamical fermions. [...]

Authors: We have moved citations both to Kogut's Review (Rev. Mod. Phys. 51, 659) and to Susskind's work on lattice fermions (Phys. Rev. D 16, 3031) to the point suggested by the Referee. Previously, these references appeared only in the Methods section.

–
Referee: Furthermore, regarding the Gauss laws. The authors neglect all the static charge sectors, besides the one with no static charges.[...]

Authors: We now explicitly state that the displayed Gauss' law specifically represents the scenario in the absence of static (background) charges.

–
Referee: The list [21-28] of one dimensional works lacks some important works of several authors, including some of the manuscript's authors as well. The choice of these particular works to be cited here (and several others only later in the context of truncating the Hilbert space in Appendix A - 51, 52, 54, 55, 57 and 59) is not entirely clear to me.

Authors: Our previous version of the manuscript was meant for Nature Physics, whose policy of limiting references to 50 in the main text forced a difficult choice on us. As we are currently submitting to Nature Communications (limit of 70, instead) we could move several of the methods refs (originally 51-59) in the main text.

–
Referee: There are many other works which could be equally mentioned [...]

Authors: All the works explicitly mentioned by the referee are now included.

–
Referee: Furthermore, at least work 23, by the way, represents a rather high dimensional scenario [...]

Authors: That reference (originally 23) has been appropriately relocated.

–
Referee: As for the works in more than one space dimension, which are more relevant for the manuscript in discussion, the list [29-33] represents very well all the collaborations and groups working in the field (with the addition of cited with the one dimensional works). However, some are missing [...]

Authors: All the works explicitly mentioned by the referee are now included.

–
Referee: Regarding the quantum simulation/computation works cited. [...]

Authors: All the works explicitly mentioned by the referee are now included, either in the main text or in the methods section (currently named appendix): We are still trying

to respect the limit of 70 refs from the main text.

—
Referee: There are also other "quantum based" methods [...]

Authors: The reference has been included.

—
Referee: In equation (A2), is the minus sign before the zero ket at the last equation required?

Authors: That minus sign was a typo, and incompatible with equation (A5). Now we rewrote (A1) and (A2) to be perfectly compatible with (A5).

—
Referee: Last paragraph of the first column of page 10: LTG should be LGT?

Authors: Yes. Corrected.

—

————— Reply to Reviewer 3: —————

We kindly thank Referee 3 for sharing our opinion that "this is the first time that this type of calculation is carried in 3+1 dimensions for a lattice gauge theories at finite density and this is a very interesting new direction of research" and for recommending publication of our manuscript. Referee 3 raised insightful questions on our results. By addressing these questions and comments we think we improved the quality of the manuscript and we made the current version easier to read than the original one.

—
Referee: The staggered fermions that are used in numerical Lagrangian calculations in 3+1 dimensions have signs that appear in the Dirac operator and discrete symmetries associated with rotations and shifts (see, for instance, Kilcup and Sharpe, Nuclear Physics B283, 493 section 2). I don't see these sign factors in the Hamiltonian of Eq. (1a). Are the authors using a different type of staggered fermions?

Authors: To our knowledge and calculations, when working in lattice Hamiltonian formulation, removing the phases from the hopping terms into all real all positive, does not affect the ground state properties of insulating phases as the ones analyzed in this work. Also, we adopt the standard Hamiltonian used, for instance, in other important works in the field of quantum simulation of LGTs, such as New J. Phys. 20, 093001 (2018). Finally, including such phases is straightforward and requires only repeating the calculations with the same procedures and algorithms presented in this manuscript.

—
Referee: The numerical calculations can be done for independent (t, m, g_e, g_m) , however the author refer to a "physical regime" where $g_e g_m = 2\sqrt{2}t$. What is the physics? QED as we know it or some condensed matter model, or ...?

Authors: With “physical regime” we refer to the original Hamiltonian formulation of lattice standard QED (Phys. Rev. D 11, 395), where these couplings are mutually related as $t = 1/a$, $m = m_0$, $g_e^2 = g^2/a$, $g_m^2 = 8/(g^2a)$, in which a is the lattice spacing, m_0 is the bare fermion mass and g is the coupling constant of QED. We added a reference related to this point in the main text (Sec. I).

–

Referee: The authors call g , the QED coupling. The dimension of g is a bit obscure. In Eq. (4), g^2 has the dimension of an energy/distance, in Eq.(5) the dimension of an energy. In continuous QED in 3+1 dimension, the coupling is usually dimensionless. Please clarify.

Authors: We agree with the Referee that equations (4) and (5) were improperly dimensioned in the previous version of the manuscript. We implicitly intended r as a lattice-distance (dimensionless integer) and g^2 should have been g_e^2 . We thank the Referee for spotting the sloppy format.

In the current version, all quantities are properly dimensioned as to be dimensionally compatible with the Hamiltonian from Eq.(1). In this version r is intended as an actual distance, as intuition suggests. Therefore, we think this version is much clearer and consistent with the other equations in the manuscript.

–

Referee: In section III (calculation with the capacitor), the authors introduce large chemical potentials on the plates. In section V (finite density) they “inject” the charges which end up on the boundary. It is not clear how this is accomplished. Would it be possible to use a uniform chemical potential to create a uniform charge density? Such calculations have been performed in 3+1 compact QED with a large mass (Kniely and Gattringer, arXiv 1502.00788).

Authors: In our calculation with the capacitor (section III), we simply added to our Hamiltonian of Eq. (1) large chemical potential local terms ($-\mu\psi_x^\dagger\psi_x$ with large positive μ) on all the sites of the two plates with opposite parity. In this way, the ground state properly reproduce the static charge distribution of a plane capacitor, with static positive/negative charges localised on the two opposite plates. While these static charges are completely “frozen” due to the large chemical potential applied, the properties of the dynamical charges in the bulk of the capacitor are completely governed by the Hamiltonian operator. Let us stress that the ground state of the capacitor always belongs to the sector of the Hilbert space with total charge equal to zero, namely perfect balance between positive and negative charges.

In the case of finite density (section V), on the contrary, we are in sectors of the Hilbert space characterized by charge imbalance into the system. In this case, we do not use any chemical potential term: as described at the beginning of the section and in Appendix B, our TTN algorithm takes into account the conservation of the total charge through the definition of global U(1) symmetry sectors encoded in the tensor networks (for a detailed and exhaustive description of the subject, please see the technical paper SciPost Phys. Lect. Notes 8 (2019)). Thus, the total charge is a quantum number that we can set as

parameter at the beginning of our simulations. As a result, we can simulate any desired charge imbalance into the system, reproducing the finite-density scenario without adding any extra-term to the Hamiltonian.

Regarding the uniform charge density states in presence of uniform chemical potential (such as the ones in Kniely and Gattlinger, arXiv 1502.00788), it would be surely possible to simulate them with our approach. In practice, it could be done by removing the aforementioned global U(1) symmetry, leaving free the total charge quantum number during the variational optimization, and by adding constant chemical potential terms on all the sites of the lattice. Similarly, it could also be done by working at fixed number of particles, determining the chemical potential as $\mu = \partial E(N)/\partial N$ and finally obtaining the grand canonical energy (at $T = 0$) as $\Omega(\mu) = E(N) - \mu N$. However, at least in the limit of large chemical potential, this situation should be very similar to what we obtained in the limit of large negative m , where the density of particles is maximum on the lattice (Figs. 2d - 2e). In our framework, the maximum density is one since we consider one species of fermion, whereas in arXiv 1502.00788 they consider two species.

–

Referee: Does the crystal phase have known physical realizations? Is the universality class of the second order phase transition to this phase known?

Authors: To the best of our knowledge, the crystal phase that we study in the present manuscript does not have physical realisations (at least for now). However, the rapid developments in low-temperature physics and control techniques could lead to a physical implementation in a very near future. For lower-dimensional LGTs, there already are several proposed and realised experimental implementations on quantum simulators based on trapped ions, ultracold atoms and Rydberg atoms in optical lattices (see Eur. Phys. J. D 74, 165 (2020) for a comprehensive description of the subject).

Regarding the universality class of the second order phase transition, we did not find in the literature any reference to its scaling-properties and universality class. For the lower dimensional analogous of our model, i.e. the 1+1 QED (Schwinger model), a similar phase transition is well known in the literature and the universality class is found to be 2D Ising with critical exponents $\beta = 1/8$ and $\nu = 1$. (see, for instance, Nuclear Physics B - Proceedings Supplements, Volume 109, Issue 1 (2002), Phys. Rev. Lett. 112, 201601 (2014)).

–

Referee: Is there any way to test the computational methods with independent numerical results? For instance, there are calculations of fermion condensates (without chemical potential) in 3+1 compact QED by Kogut and collaborators (PRL 61, 2416 and later publications).

Authors: As suggested by the Referee, we carried out additional calculations for testing our approach with other independent numerical results and to prove the validity of our findings. In particular, we focused on the computation of the chiral condensate at zero fermion mass in order to benchmark our results with the ones presented in PRL 61, 2416

(the paper pointed out by the Referee). Here we report all the steps of our analysis and

Figure 1: Computation of the Chiral Condensate

we explain the results shown in Fig. 1:

- (i) the chiral fermion condensate, on a first approximation, $\langle \bar{\psi}\psi \rangle$ can be computed as the first derivative of the energy-density with respect to the bare fermion mass (Phys.Lett.B336:290-294,1994, Phys. Rev. D 93, 094512, Phys. Rev. D 101, 054507), namely, taking into account a three-dimensional lattice of linear size L , $\langle \bar{\psi}\psi \rangle \approx \frac{1}{L^3} \frac{\partial E}{\partial m}$;
- (ii) we consider two different values for the coupling g , so that $\beta = 1/g^2 = 0.20$ and $\beta = 0.33$. In PRL 61, 2416, in correspondence of these values for β , the two different regimes for the scaling of the chiral condensate are clearly visible. For each of these values of β , we simulate our model for different values of the mass m in the range $[-0.5, 0.5]$ and for the three linear sizes $L = 2, 4, 8$, determining the ground state energy-densities as shown in Fig. 1 (first row panels);

- (iii) we numerically compute the first derivative of these energy-densities, as reported in Fig. 1 (second row panels), highlighting different behaviors of this quantity close to the point $m = 0$. In particular, we observe that for $\beta = 0.20$, it increases with the size of the lattice, whereas for $\beta = 0.33$, the numerical values of the derivative are smaller and tend to decrease with L .
- (v) we consider the values of the derivatives for $m = 0$ and we analyse their scaling as a function of the system size, as shown in Fig. 1 (third row panels), highlighting an approximate linear scaling. By fitting these data, we extract a rough estimate for the value of the chiral condensate in the limit $m \rightarrow 0$ and $L \rightarrow \infty$. We get $\langle \bar{\psi}\psi \rangle_{m=0} \approx 0.397$ for $\beta = 0.20$ and $\langle \bar{\psi}\psi \rangle_{m=0} \approx 0.095$ for $\beta = 0.33$.

In PRL 61, 2416 are not reported the values of the chiral condensate for $N_f = 1$ (where N_f is the number of species of dynamical fermions), that is the case of our model. However, the results we obtain are compatible with the general scaling behavior, but above all they show the presence of two different regimes as a function of β , as expected from the findings in PRL 61, 2416. We stress that our model contains several approximations, such as the discretization of the gauge degrees of freedom through the finite quantum-link representation $s = 1$, and that, for a precise and fully meaningful numerical comparison, we should consider the large- s limit and also the limit in which the lattice spacing a goes to zero. In principle these analysis would be definitely fitting for TTN methods, with no formal or theoretical limitations. However, they are extremely time-consuming (for user- and computer run-time) and thus are well beyond the scope of the current manuscript. In particular, from the computational point of view, we would like to point out that we have not exploited yet, in our TTN algorithms, the power of a massive parallelization on multi-node (e.g. MPI library) or GPU architectures (these computational setup are at the basis of current Monte Carlo simulations of LGTs after years of technical development; see for instance Phys. Rev. D 98, 030001). We are not at that stage yet, but we know how to scale our algorithms following these paradigms and we are already working in this direction with some preliminary tests. In conclusion, we are extremely confident that we can address the aforementioned steps towards the continuum limit in a near future and that all the work presented in this manuscript forms a strong basis to stimulate further research in this direction.

–

Referee: Minor presentation details:

-The last sentence of the abstract says that fundamental questions are addressed. It would be good to say what the findings are.

-First paragraph of the introduction: ...(QCD), a founding pillar ...for almost a century. I believe the authors mean "half a century" (or I misunderstood the sentence).

Authors:

- We extended the last sentence of the abstract with "we address fundamental questions such as the characterization of collective phases of the model, the presence of a confining

phase at large gauge coupling, and the study of charge-screening effects”.

-The Referee is correct. We replaced “for almost a century” with “for almost half a century” in the Introduction.

Reviewers' Comments:

Reviewer #1:

Remarks to the Author:

The authors have presented some plausible arguments in favor of their work. I agree with them that their paper is a very interesting development and there are good reasons for following their line of research. However, I am still of the opinion that the current version is making a somehow too enthusiastic impression. This is certainly positive that the authors are enthusiastic about their own work, but I would still vote for a bit more critical discussion of the challenges that have to be faced before eventually a meaningful application e.g. to QCD can emerge. I think such a discussion would profit the reader who is really interested in following this line of research and understanding the crucial challenges.

First, I would like to come back to the entanglement scaling issue. I fully agree with the authors that higher-dimensional systems will likely require comparatively moderate bond dimensions and this is a positive prospect. Still, I think it would be extremely valuable to show the convergence in bond dimension for a couple of system sizes - I mean Fig.8c for other values of L . That would provide more intuition how important is the issue of the scaling of the bond dimension with the system size. $L=8$ is still extremely small when having QCD in mind and the authors should clearly point out (or justify otherwise) that improvements of the simple TTN Ansatz are needed to tackle realistic system sizes. The fact that this is "just" a technical challenge is clear, but I believe it will be a non-trivial technical issue and one that the readers should be aware of. In other words, is it realistic to have simulations converged in bond dimension e.g. for $L=64$? Showing the increase of the required bond dimension for the considered small system sizes would shed some light on this and my intuition is that aTTN or another improvement over simple TTN is a must for the future, which would be an important conclusion for future work.

Second, I think the challenge of including higher-spin representations should also be highlighted. Having now only the lowest non-trivial representation leads to a local dimension of 267. What would it be if including also the next one? Also, do the authors know the corresponding numbers for an $SU(2)$ gauge theory and for QCD? One can argue this just a "technicality", but I am convinced that in practice it will also be a huge challenge, already for QED and let alone for QCD.

Summarizing, the authors have good arguments why this direction of work is prospective and I agree that it can lead to a lot of important follow-up work, both from the authors and hopefully also from other groups. Thus, the only missing element from my perspective is that the authors should be more outspoken about the challenges that are still to be faced and will be very non-trivial to handle having robust applications to real-world theories in mind.

Reviewer #3:

Remarks to the Author:

The authors have addressed most of the criticisms in a satisfactory manner. However there are still two points that require adjustments.

The first one is the statement in 2011.08200 by some of the authors that the "TTN fail to capture the area law for large systems." Referee 1 explains clearly that more modest conclusions should be drawn given this statement. I don't think that the modified paragraph in the outlook conveys this message.

The second concerns the physics applications of the model calculations. The staggered signs that, as pointed out by referee 3, should appear in eq. (1a) come from a specific representation of the gamma matrices and are crucial to describe some features of Dirac fermions in the continuum limit. Also invoking Phys. Rev. D 11 for the continuum limit of lattice QED does not seem correct because this article deals with the non-abelian, asymptotically free, case for which the continuum limit is completely different than QED. The continuum limit of compact lattice QED is a difficult question. I think the statement by referee 1 that "While the theory sounds akin to real-world

QED, its properties are pretty much different ..." is accurate and should be acknowledged at the beginning of the article.

————— **Reply to Reviewer 1:** —————

We warmly thank Referee 1 for stating that “the authors have presented some plausible arguments in favor of their work. I agree with them that their paper is a very interesting development and there are good reasons for following their line of research” and that “the authors have good arguments why this direction of work is prospective and I agree that it can lead to a lot of important follow-up work, both from the authors and hopefully also from other groups.”

—

Referee: The authors have presented some plausible arguments in favor of their work. I agree with them that their paper is a very interesting development and there are good reasons for following their line of research. However, I am still of the opinion that the current version is making a somehow too enthusiastic impression. This is certainly positive that the authors are enthusiastic about their own work, but I would still vote for a bit more critical discussion of the challenges that have to be faced before eventually a meaningful application e.g. to QCD can emerge. I think such a discussion would profit the reader who is really interested in following this line of research and understanding the crucial challenges.

Authors: Following Referee 1 suggestion, we extended the Discussion section, listing from a feasibility perspective all the steps and the possible strategies that one can follow in the near future to face the important challenge of extending the presented TTN approach to more complicated and realistic gauge theories. We agree with Referee 1 that this discussion can be of great interest to the reader and we believe that the new version has improved a lot, also thanks to these comments.

—

Referee: First, I would like to come back to the entanglement scaling issue. I fully agree with the authors that higher-dimensional systems will likely require comparatively moderate bond dimensions and this is a positive prospect. Still, I think it would be extremely valuable to show the convergence in bond dimension for a couple of system sizes - I mean Fig.8c for other values of L . That would provide more intuition how important is the issue of the scaling of the bond dimension with the system size. $L=8$ is still extremely small when having QCD in mind and the authors should clearly point out (or justify otherwise) that improvements of the simple TTN Ansatz are needed to tackle realistic system sizes. The fact that this is “just” a technical challenge is clear, but I believe it will be a non-trivial technical issue and one that the readers should be aware of. In other words, is it realistic to have simulations converged in bond dimension e.g. for $L=64$? Showing the increase of the required bond dimension for the considered small system sizes would shed some light on this and my intuition is that a TTN or another improvement over simple TTN is a must for the future, which would be an important conclusion for future work.

Authors: Following Referee 1 suggestion, Fig.8c has been modified in order to show the convergence in bond dimension for all the system sizes that we considered. It is possible to see that, although the value of the bond dimension needed for reaching the convergence

plateau increases a bit with the sizes, it remains fairly moderate, denoting that the 3D model actually shows a low or intermediate entanglement content that TTNs are able to encode for the limited sizes presented in this work. However, we agree that for very large sizes (such as $L = 64$), the aTTNs in combination with a massive-parallelization could be the right way to tackle the problem of the “continuum limit”, as now explained in the Discussion section. We are already working in this direction with some preliminary tests, trying also to implement and test optimization strategies for truncating the local Hilbert space basis of higher-spin representations of the gauge fields (see for instance arXiv:2011.07412). This would be important especially for aTTN since they require the optimization of the disentanglers that are d^4 tensors, where d is the local basis dimension. Usually, in the context of spin or fermionic models, d is extremely small comparative to the reasonably reachable bond dimensions m and, thus, the leading term of the algorithmic scaling of aTTN remains $O(m^4 d^2)$, as described in arXiv:2011.08200. On the contrary, in the context of high-dimensional LGTs, d can be easily huge and optimization strategies of the local basis could be required to allow for efficient simulations with aTTN.

–

Referee: Second, I think the challenge of including higher-spin representations should also be highlighted. Having now only the lowest non-trivial representation leads to a local dimension of 267. What would it be if including also the next one? Also, do the authors know the corresponding numbers for an SU(2) gauge theory and for QCD? One can argue this just a “technicality”, but I am convinced that in practice it will also be a huge challenge, already for QED and let alone for QCD.

Authors: Following Referee 1 comments, we included these points in the Discussion section, adding also the expected local basis dimensions for the next spin representation of the gauge fields of QED and for the non-Abelian SU(2) Yang-Mills theory. We also highlighted the steps that we believe are needed and in principle feasible in the context of our TTN approach for tackling the problem of these higher-spin representations.

–

Referee: Summarizing, the authors have good arguments why this direction of work is prospective and I agree that it can lead to a lot of important follow-up work, both from the authors and hopefully also from other groups. Thus, the only missing element from my perspective is that the authors should be more outspoken about the challenges that are still to be faced and will be very non-trivial to handle having robust applications to real-world theories in mind.

Authors: We hope that Referee 1 could now appreciate the new paragraphs in the Discussion section, focused exactly on the description of the challenges that are still to be faced for extending the TTN approach to real-world LGTs. We thank the referee for the previous comments that stimulated us to improve the manuscript with respect to our first version.

————— **Reply to Reviewer 3:** —————

We are happy that Referee 3 is satisfied with our previous reply. We welcomed the new comments of Referee 3, which led us to improve the Discussion section.

—
Referee: The first one is the statement in 2011.08200 by some of the authors that the "TTN fail to capture the area law for large systems." Referee 1 explains clearly that more modest conclusions should be drawn given this statement. I don't think that the modified paragraph in the outlook conveys this message.

Authors: Following Referee 3 comments, we substantially expanded the Discussion section, by adding new paragraphs concerning this point.

—
Referee: The second concerns the physics applications of the model calculations. The staggered signs that, as pointed out by referee 3, should appear in eq. (1a) come from a specific representation of the gamma matrices and are crucial to describe some features of Dirac fermions in the continuum limit. Also invoking Phys. Rev. D 11 for the continuum limit of lattice QED does not seem correct because this article deals with the non-abelian, asymptotically free, case for which the continuum limit is completely different than QED. The continuum limit of compact lattice QED is a difficult question. I think the statement by referee 1 that "While the theory sounds akin to real-world QED, its properties are pretty much different ..." is accurate and should be acknowledged at the beginning of the article.

Authors: We agree that the continuum limit of lattice QED is a difficult question, and our work intentionally does not delve deep in this direction. As such, we have reformulated the main sentence of our contribution: We now state that we are simulating an abelian lattice theory akin to lattice QED.

Reviewers' Comments:

Reviewer #1:

Remarks to the Author:

The authors have included a comprehensive discussion on the challenges of future research directions. I think now the readers are offered a good overview of the current situation. Thus, they can appreciate the very interesting application of TTN first time to a 3+1-dim lattice gauge theory, but at the same time be aware that it does not immediately mean that full QCD is within reach very soon. I still believe many of the challenges are highly non-trivial, at least at the practical level, but the authors' work can clearly invoke a lot of interesting developments. Hence, I believe it should now be published.

————— **Reply to Reviewer 1:** —————

Referee: The authors have included a comprehensive discussion on the challenges of future research directions. I think now the readers are offered a good overview of the current situation. Thus, they can appreciate the very interesting application of TTN first time to a 3+1-dim lattice gauge theory, but at the same time be aware that it does not immediately mean that full QCD is within reach very soon. I still believe many of the challenges are highly non-trivial, at least at the practical level, but the authors' work can clearly invoke a lot of interesting developments. Hence, I believe it should now be published.

Authors: We warmly thank the referee for his recommendation and for the previous comments that stimulated us to improve the manuscript with respect to our first version.